# Mitochondrial respiration controls neoangiogenesis during wound healing and tumour growth

L. M. Schiffmann[1,2,3,10], J. P. Werthenbach[1,10], F. Heintges-Kleinhofer[1], J. M. Seeger[1], M. Fritsch[1], S. D. Günther[1], S. Willenborg[4,5], S. Brodesser [6], C. Lucas[6], C. Jüngst [7], M. C. Albert[1], F. Schorn[1], A. Witt[1], C. T. Moraes[8], C. J. Bruns[2,3], M. Pasparakis [9], M. Krönke[1], S. A. Eming[4,5], O. Coutelle[1] & H. Kashkar [1,4✉]

The vasculature represents a highly plastic compartment, capable of switching from a quiescent to an active proliferative state during angiogenesis. Metabolic reprogramming in endothelial cells (ECs) thereby is crucial to cover the increasing cellular energy demand under growth conditions. Here we assess the impact of mitochondrial bioenergetics on neovascularisation, by deleting *cox10* gene encoding an assembly factor of cytochrome c oxidase (COX) specifically in mouse ECs, providing a model for vasculature-restricted respiratory deficiency. We show that EC-specific *cox10* ablation results in deficient vascular development causing embryonic lethality. In adult mice induction of EC-specific *cox10* gene deletion produces no overt phenotype. However, the angiogenic capacity of COX-deficient ECs is severely compromised under energetically demanding conditions, as revealed by significantly delayed wound-healing and impaired tumour growth. We provide genetic evidence for a requirement of mitochondrial respiration in vascular endothelial cells for neoangiogenesis during development, tissue repair and cancer.

---

[1] Institute for Medical Microbiology, Immunology and Hygiene (IMMIH), Cologne Excellence Cluster on Cellular Stress Responses in Aging-Associated Diseases (CECAD), Faculty of Medicine and University Hospital of Cologne, University of Cologne, Cologne, Germany. [2] Department of General, Visceral, Cancer and Transplant Surgery, Faculty of Medicine and University Hospital of Cologne, University of Cologne, Cologne, Germany. [3] Centre for Integrated Oncology (CIO) Cologne Bonn, Cologne, Germany. [4] Centre for Molecular Medicine Cologne (CMMC), Faculty of Medicine and University Hospital of Cologne, University of Cologne, Cologne, Germany. [5] Department of Dermatology, Cologne Excellence Cluster on Cellular Stress Responses in Aging-Associated Diseases (CECAD), Faculty of Medicine and University Hospital of Cologne, University of Cologne, Cologne, Germany. [6] Lipidomics/Metabolomics Facility, Cologne Excellence Cluster on Cellular Stress Responses in Aging-Associated Diseases (CECAD), University of Cologne, Cologne, Germany. [7] Imaging Facility, Cologne Excellence Cluster on Cellular Stress Responses in Aging-Associated Diseases (CECAD), University of Cologne, Cologne, Germany. [8] Department of Neurology, Miller School of Medicine Miami, University of Miami, Miami, Florida, USA. [9] Institute for Genetics, Cologne Excellence Cluster on Cellular Stress Responses in Aging-Associated Diseases (CECAD), Centre for Molecular Medicine Cologne (CMMC), University of Cologne, Cologne, Germany. [10] These authors contributed equally: L. M. Schiffmann, J. P. Werthenbach. ✉email: h.kashkar@uni-koeln.de

Blood vessels, while mostly remaining quiescent throughout adult life, possess the capacity to rapidly form new sprouts (neoangiogenesis) in response to injury or under pathological conditions requiring a blood supply. The inner surface of blood vessels is lined by a thin layer of endothelial cells (ECs) with remarkable functional plasticity that can rapidly switch from a quiescent to a highly proliferative state required for neoangiogenesis[1]. ECs have long been considered to be metabolically inert and the angiogenic switch was thought to be solely regulated by growth factor signalling[2]. However, recent evidence indicates that EC metabolism is intimately linked to EC function and even controls EC-fate decisions during angiogenesis[3,4]. Accordingly, besides glycolysis, fatty acid oxidation (FAO)[5] and glutamine metabolism[6,7] are now also recognised as metabolic pathways that significantly contribute to angiogenesis.

Glucose is regarded as the body's most readily available source of energy. Under anaerobic conditions glucose is first converted to pyruvate and then to lactic acid in the cytoplasm of cells undergoing glycolysis. Loss of endothelial PFKFB3—an indirect activator of glycolysis—results in reduced vessel formation in mice providing genetic evidence for the pivotal role of glycolysis in sprouting angiogenesis[3]. In the presence of oxygen, pyruvate may be used for oxidative phosphorylation (OxPhos) in the mitochondria with a much higher ATP yield per glucose than glycolysis. Indeed, oxygen does not appear to be a limiting factor as ECs typically have access to the high intravascular oxygen environment. However, the role of OxPhos in EC function remains controversial[4]. For example, reactive oxygen species (ROS) generated by OxPhos may lead to endothelial dysfunction while glycolysis is thought to fuel the production of metabolic intermediates required for sustained growth. Nevertheless, inhibitors of mitochondrial respiration effectively inhibit tumour growth by disrupting tumour blood vessels[8–10]. A recent study showed that inhibition of respiratory chain complex III impairs EC proliferation and angiogenesis suggesting that the primary role of mitochondria in ECs is to serve as biosynthetic organelles for cell proliferation[11]. These observations suggest that OxPhos could have a more critical role in EC function than currently appreciated.

To selectively assess the role of OxPhos for EC function, we established a mouse line with an EC specific deficiency of cytochrome c oxidase (COX, Complex IV) by deleting the *cox10* gene —a protoheme:heme-O-farnesyl transferase required for the synthesis of heme a, the prosthetic group of the catalytic centre of COX[12]. Missense mutations in *cox10* are associated with various human disorders[12,13] and tissue-specific *cox10* ablation in mouse muscle or liver results in progressive mitochondrial myopathy[12] or mitochondrial hepatopathy[13], respectively. Here we show that EC-specific knockout (KO) of *cox10* in mice results in embryonic lethality. Furthermore, loss of endothelial OxPhos in adult mice slows wound vascularisation and healing and reduces tumour growth and angiogenesis. In contrast, under homeostatic conditions, EC-specific *cox10*-deficiency, produces no significant phenotype. Our data provide genetic evidence for the requirement of OxPhos during angiogenesis and shed new light on the metabolic activity of ECs during tumour growth and development.

## Results

**Endothelial COX10 is essential for embryonic development**. To explore the role of OxPhos in vascular development we specifically deleted *cox10* in the mouse endothelium by cross-breeding mice carrying loxP-flanked *cox10* alleles (*cox10^{fl/fl}*)[12] with *Tie2-Cre* transgenic mice, in which expression of Cre recombinase is driven by an EC-specific promoter/enhancer[14] (Fig. 1a, Supplementary

Fig. 1a–f). Out of 76 progenies from *Tie2-Cre+cox10^{fl/wt}* × *cox10^{fl/fl}* mouse intercrosses, not one was born with a homozygous deletion of endothelial *cox10* (*cox10^{EC−/−}*), indicating that *cox10*-deficiency in ECs resulted in embryonic lethality (Fig. 1a, b and Supplementary Fig. 1a, b).

Macroscopically, *cox10^{EC−/−}* embryos appeared pale at embryonic day 10.5 (E10.5) compared with wild-type and *cox10^{EC−/+}* embryos, suggesting reduced vascular development (Supplementary Fig. 1b). These differences became progressively more evident at later time points of embryonic development (by day E12.5) (Fig. 1b, c) and ultimately resulted in developmental impairment and death of *cox10^{EC−/−}* embryos. Besides its expression in ECs, Tie2 is also expressed in around 80–90% of hematopoietic cells including lymphocytes and macrophages[15,16]. However, genetic disruption of mitochondrial respiration in lymphocytes did not result in embryonic lethality[17–19]. Similarly, we did not observe embryonic lethality upon specific ablation of *cox10* in myeloid cells using *LysM-Cre* transgenic mice[20] (Supplementary Fig. 1g-h). Furthermore, a recent study using *Vav-iCre* mice targeting mitochondrial respiration mainly in hematopoietic stem cells (HSCs) but also in ECs[21] displayed normal Mendelian ratios at E15.5 resulting in embryonic lethality after E18.5[22]. Together these observations suggest that embryonic lethality observed in *cox10^{EC−/−}* embryos at E12.5 is unlikely due to the off-target effects of COX deficiency in tissues other than ECs. Accordingly, yolk sac analysis not only revealed a reduction in vascular density in *cox10^{EC−/−}* embryos, but clearly showed vascular defects (Fig. 1c–e and Supplementary Fig. 1e). Yolk sacs from *cox10^{EC−/−}* at E12.5 completely lacked the typical vascular hierarchy with multiple blind ends (Fig. 1c, red arrows, Fig. 1e) suggestive of a developmental arrest at the stage of the primordial plexus. Whole-mount three-dimensional imaging of embryonic vascular network (Fig. 1f, g and Supplementary Movies 1–6) also revealed severe defects in vascularity. Together, these data clearly indicate that *cox10* KO in ECs results in impaired vascular development and embryonic lethality in mice.

For detailed metabolic characterisation of ECs lacking OxPhos we isolated ECs from *cox10^{fl/fl}* mice and induced *cox10* gene deletion in vitro using a cell permeable active Cre protein[23]. The generated *cox10*-deficient cell line is hereafter referred to as *cox10^{KO}*. Purity of the isolated cells and efficient in vitro deletion of the *cox10* gene were first verified (Supplementary Fig. 2a–e) and the respiratory capacity of the isolated ECs was assessed with a Seahorse extracellular flux (XF) analyser (Fig. 2a–e). As expected, the oxygen consumption rate (OCR) was drastically reduced in *cox10*-deficient ECs with a significant reduction in the calculated basal and spare respiratory capacities (Fig. 2a–c). Furthermore, the fraction of basal respiration that was being used to drive ATP production as well as the remaining basal respiration that was not coupled to ATP production (proton leak) were both significantly reduced (Fig. 2d, e), consistent with OxPhos dysfunction in the *cox10^{KO}* cells. The reduced respiratory capacity of *cox10*-deficient ECs was accompanied by a compensatory increase in glycolysis (Fig. 2f–h). After addition of saturating amounts of glucose, the extracellular acidification rate (ECAR) was boosted to the maximum glycolytic capacity in *cox10^{KO}* ECs. Comparable levels of glycolysis are only observed in wild-type ECs in the presence of oligomycin, which effectively shuts down OxPhos thus driving the cells to use glycolysis to its maximum capacity (Fig. 2f). Depending on the availability of nutrients, and glucose in particular, only few tissues possess the ability to switch between glycolysis and OxPhos, to adapt their metabolism to the prevailing environmental conditions. Indeed, increasing concentrations of glucose gradually increased ECAR and reduced OCR in murine ECs and human umbilical vein ECs (HUVECs) (Fig. 2i and Supplementary Fig. 2f). Thus, under

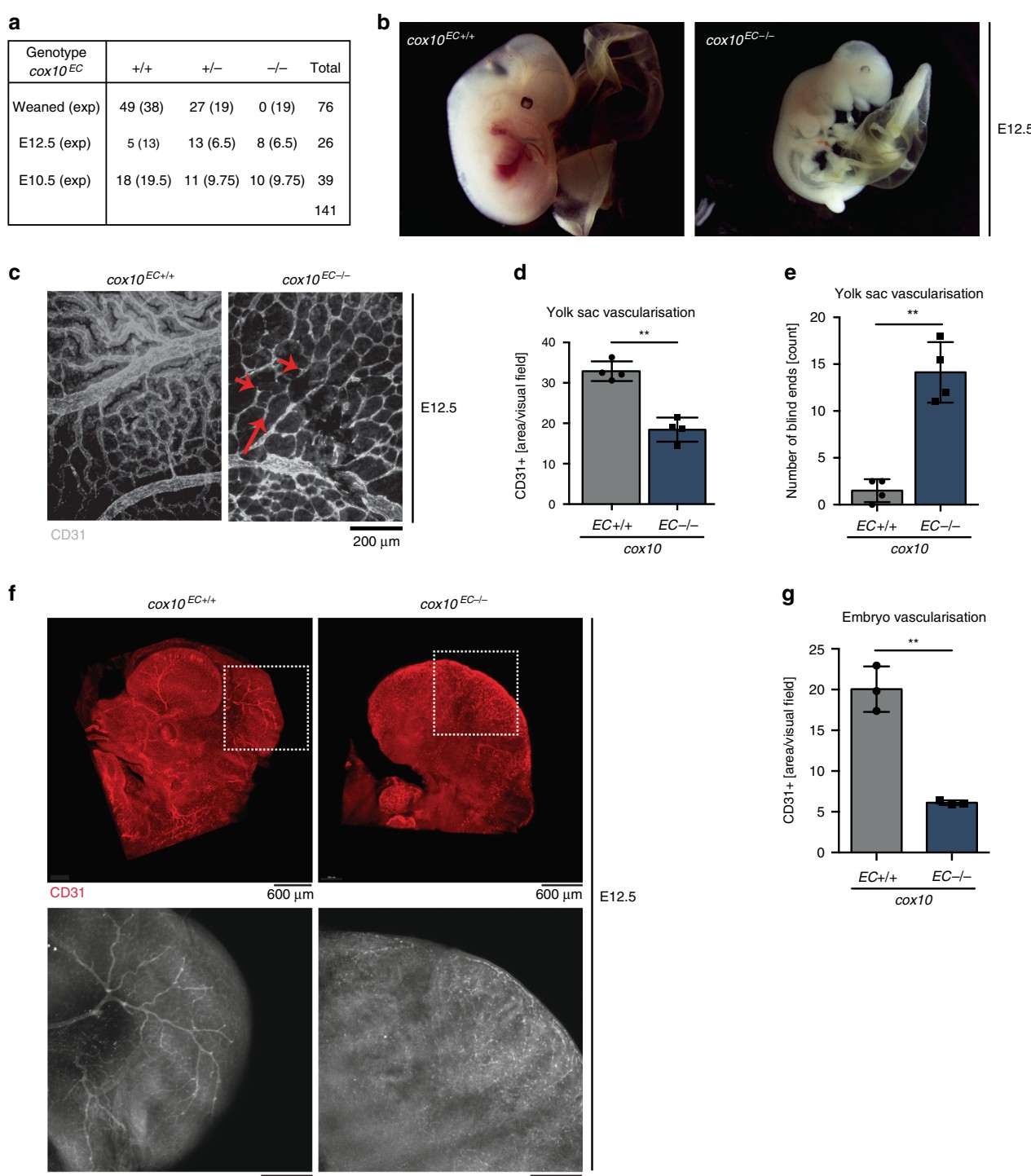

**Fig. 1 Endothelial *cox10* is required for embryonic development. a** Genotypes of weaned mice or embryos at different stages of gestation obtained from crossing Tie2-Cre *cox10<sup>fl/wt</sup>* to *cox10<sup>fl/fl</sup>*. **b** Representative images of *cox10<sup>EC+/+</sup>* (*cox10<sup>fl/fl</sup>*) and *cox10<sup>EC−/−</sup>* E12.5 embryos. **c** Whole-mount yolk sacs stained with anti-CD31 as an endothelial marker and **d** quantification of CD31+ yolk sac vasculature (*n* = 4/genotype). **e** Quantification of blind ends (*n* = 4/genotype). **f** Whole embryos (E12.5) stained with anti-CD31 in 5-fold (upper panel) and 10-fold (lower panel) magnification. **g** Quantification of embryo CD31 area density (*n* = 3/genotype). Data are presented as mean ± SD. Individual data points in (**d**, **e**, **g**) represent analysed mice per genotype (**d, e** *n* = 4, **g** *n* = 3). Exact *p*-values (unpaired students *t*-test, two-tailed): **d** 0.0003, **e** 0.0003, **g** 0.0010.

saturating amounts of glucose, OxPhos in *cox10*-competent ECs was suppressed. Together, these findings indicate that the mitochondrial dysfunction in COX-deficient ECs may be alleviated by the compensatory utilisation of glucose as ECs obviously have the ability to flexibly adjust their energy utilisation according to nutrient availability. Metabolic profiling revealed that COX deficiency resulted in an accumulation of TCA cycle intermediates such as fumarate, malate and succinate, and reduction of glycolytic intermediates, and, in particular, significantly decreased cellular ATP levels indicating that EC mitochondrial respiration considerably contributes to cellular energy production (Fig. 2j).

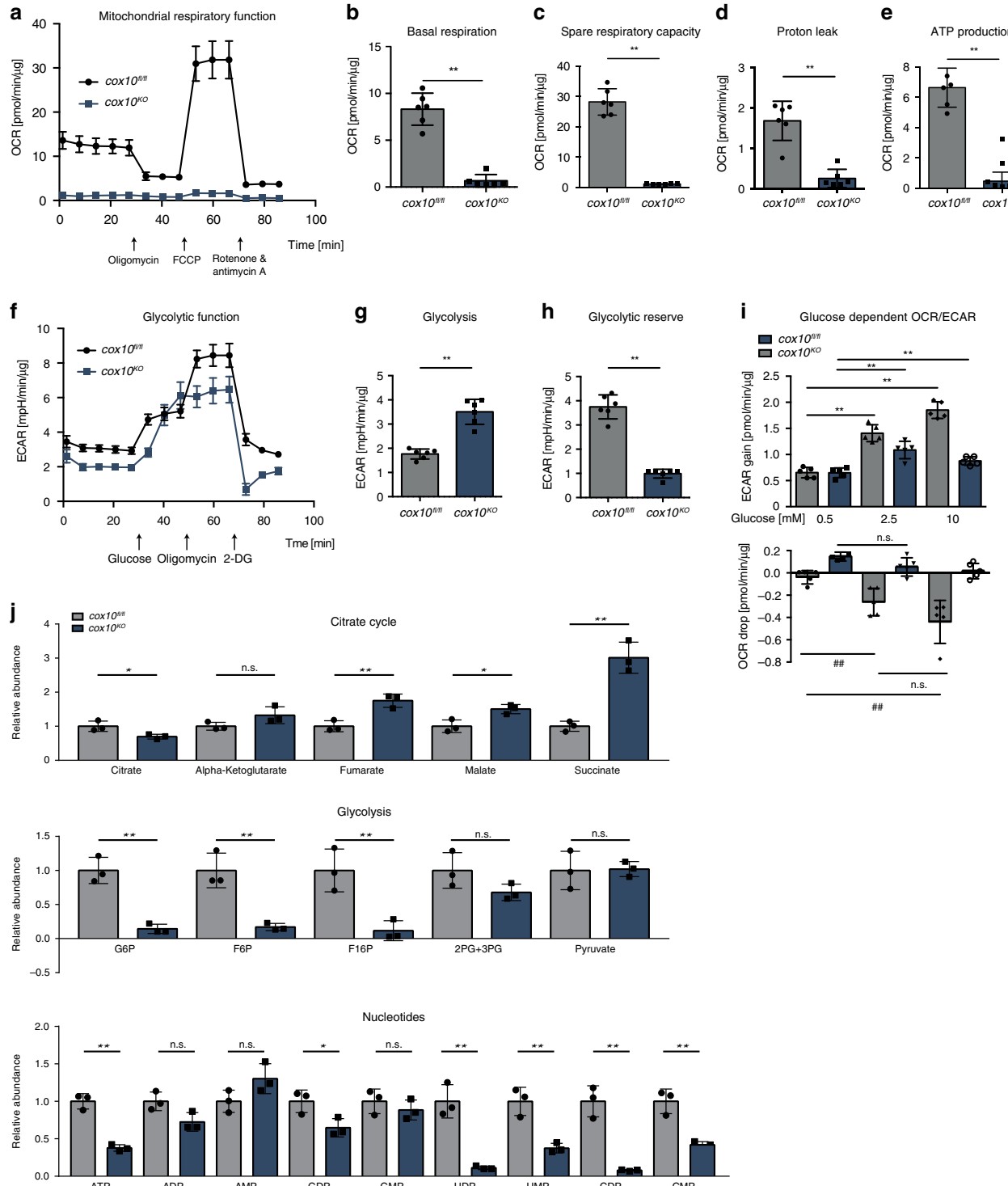

**Fig. 2 Loss of endothelial *cox10* changes EC metabolic phenotype. a** Oxygen consumption rates (OCR) of COX proficient *cox10^fl/fl* control primary ECs vs. COX-deficient *cox10^KO* ECs before and after sequential injection of oligomycin, FCCP and a mixture of rotenone/antimycin A by a Seahorse XF96 analyser. **b–e** Corresponding calculated parameters of mitochondrial respiration. **f** Extracellular acidification rate (ECAR) of *cox10^fl/fl* vs. *cox10^KO* ECs after sequential injection of glucose, oligomycin and 2-DG (2-deoxy-D-glucose). **g, h** corresponding calculated parameters expressing glycolytic capacity of ECs. **i** Calculation of drop in OCR or gain in ECAR of isolated murine ECs (*cox10^fl/fl* control primary ECs vs. COX-deficient *cox10^KO* ECs) in response to different glucose concentrations. **j** Metabolomic analysis of pathway intermediates, grouped into citrate cycle, glycolysis and nucleotides, normalised to *cox10^fl/fl* controls. Data are presented as mean ± SD. Individual data points in (**a**)–(**j**) represent technical replicates within a representative experiment, sample size: **a–h** $n = 6$, **i** $n = 5$, **j** $n = 3$. Exact $p$-values (unpaired students $t$-test, two-tailed): **b, c, d, e, g, h** <0.0001; **i** gain from bottom to top: **<0.0001, **=0.0024, **=0.0009, **=0.0034. Drop from bottom to top: ##: 0.0022, n.s.: 0.1202, ##: 0.0066, n.s.: 0.0515. **j** Citrate cycle from left to right: 0.0339, 0.1097, 0.0068, 0.0191, 0.0019; glycolysis from left to right: 0.0019, 0.0052, 0.0116, 0.1247, 0.9148. Nucleotides from left to right: 0.0006, 0.0534, 0.1033, 0.0348, 0.3952, 0.0024, 0.0055, 0.0015, 0.0041.

**Mitochondrial respiration is required for EC function**. Surprisingly, under standard cell culture conditions with physiological glucose concentrations, *cox10*-deficient ECs exhibited no obvious morphological phenotype. We therefore examined if glucose availability had an impact on the viability of COX-deficient ECs by using three independent viability assays in vitro

(Fig. 3a and Supplementary Fig. 3a, b). In the presence of physiological or hyperglycaemic glucose levels, the viability of *cox10*[fl/fl] EC was not affected. In the absence of glucose, *cox10*[fl/fl] ECs survived by engaging OxPhos to cover their energetic demand. In contrast, *cox10*[KO] ECs lacking OxPhos were highly susceptible to glucose deprivation and responded with a gradual

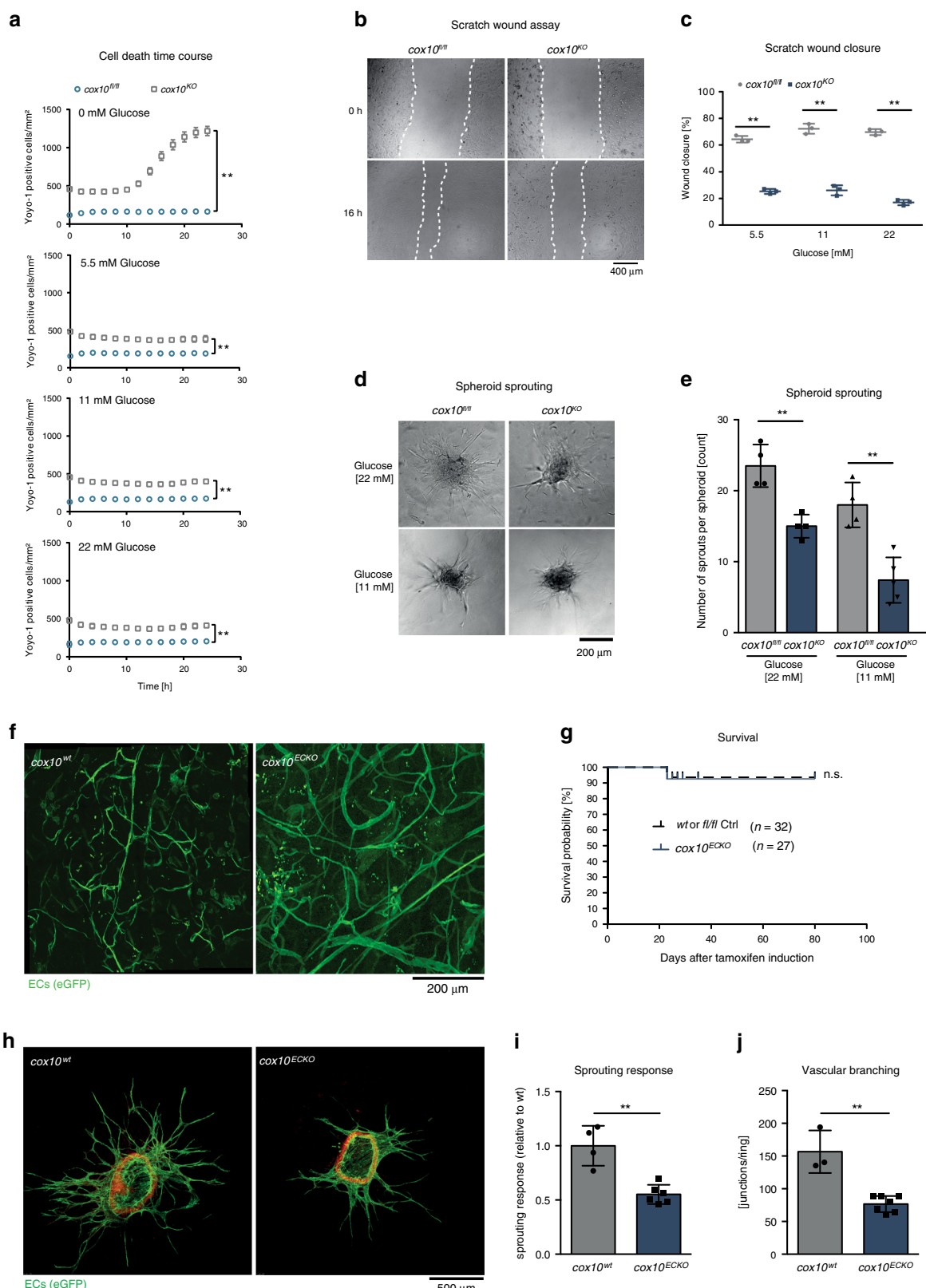

**Fig. 3 Cox10 is essential for EC function. a** Time course of cell death of $cox10^{fl/fl}$ ECs and $cox10^{KO}$ ECs in the presence of different glucose concentrations measured by yoyo-1 uptake. **b** Representative images of scratch wound assay of $cox10^{fl/fl}$ and $cox10^{KO}$ ECs after 16 h cultured in medium containing 5.5-mM glucose. **c** Quantification of scratch wound closure of monolayer cultured control $cox10^{fl/fl}$ and $cox10^{KO}$ ECs (% of wound closure) at indicated glucose levels after 16 h. **d** Representative bright field images of sprouting spheroids. **e** Quantification of sprouts per spheroid. **f** Maximum projections of eGFP-expressing ECs generated from multiphoton microscopy z-stacks of murine ear tissue of $cox10^{wt}$ (EndSCLCreERT/R26mTmG) or $cox10^{ECKO}$ (EndSCLCreERT/ $cox10^{fl/fl}$/R26mTmG) mice after tamoxifen treatment. GFP expression indicates EC-specific Cre recombination. **g** Kaplan–Meier curve showing survival probability of $cox10^{wt \ or \ fl/fl}$ mice vs. $cox10^{ECKO}$ mice. **h** Representative fluorescent images (maximum projection of confocal stacks) of aortic rings derived from $cox10^{wt}$ (EndSCLCreERT/R26mTmG) or $cox10^{ECKO}$ (EndSCLCreERT/$cox10^{fl/fl}$/R26mTmG) mice after tamoxifen treatment. **i** Quantification of GFP+ sprouts and **j** junctions. Data are presented as mean ± SD. Individual data points in (**a**), (**c**) and (**e**) represent technical replicates within a representative experiment. Sample size: **a** $n = 3$, **c** $n = 3$, **e** $n = 4$, except $cox10^{KO}$ 11 mM: $n = 5$. Individual data points in (**i**) and (**j**) represent mean values of individual mice of the respective genotype. Sample size: **i** $n = 4$ (wt) vs. $n = 6$ (ECKO), **j** $n = 3$ (wt) vs. $n = 7$ (ECKO). Exact p-values (unpaired students t-test, two-tailed): **a** 0 mM: <0.0001, 5 mM: 0.0021, 11 mM: <0.0001, 22 mM: 0.00057 (comparing time point 24 h), **e** 0.0025 (22 mM), 0.0016 (11 mM), **i** 0.0008, **j** 0.0003. Log-rank test was used to determine differences between survival curves in (**g**), exact p-value: 0.8607. P-values (one-way ANOVA followed by Bonferroni post hoc test): **c** **<0.0001.

decrease in viability (Fig. 3a and Supplementary Fig. 3a, b). In line with a previous study[11], the OxPhos inhibitors antimycin A and oligomycin did not induce significant cytotoxicity in primary murine ECs and HUVECs. In the absence of glucose, however, murine ECs and HUVECs became highly susceptible to treatment with antimycin A and oligomycin (Supplementary Fig. 3c–e) indicating that limited glucose availability led to an increasing dependency on OxPhos. Notably, glucose deprivation induced cell death in COX-deficient ECs which could not be rescued by the pan-caspase inhibitor zVAD-fmk excluding apoptosis as the main driver of EC death in this experimental system (Supplementary Fig. 3f).

Further in vitro studies revealed a significant effect of COX dysfunction on EC functionality. Specifically, 3D spheroid sprouting assays[24] and scratch wound assays showed that sprouting response and migration of $cox10^{KO}$ ECs was significantly hampered which could not be rescued by supra-physiological glucose concentrations (Fig. 3b–e). Cox10-deficiency also significantly reduced EC proliferation (Supplementary Fig. 3g) which is in line with a recent observation reporting an antimycin A (OxPhos inhibitor)-mediated proliferation defect in HUVECs[11]. Notably, the pre-treatment of cells with mitomycin C, which is commonly used to block cell proliferation in cell migration assays, did not cause a gross delay in wound closure in our experimental setup (Supplementary Fig. 3h). This suggests that the reduced migratory capacity of $cox10^{KO}$ ECs rather than their reduced proliferation underlies the EC dysfunction measured in 3D spheroid sprouting and scratch wound assays. Together, these data indicate that ECs are dependent on OxPhos to execute important cellular functions including proliferation, migration and sprouting.

**EC OxPhos is dispensable for adult vascular homeostasis.** We have shown above that endothelial COX deficiency is associated with embryonic lethality. To further explore the physiological role of OxPhos in ECs, we induced cox10 deletion in ECs of adult mice harbouring an endothelial-specific, tamoxifen-inducible Cre-transgene (EndSCL-Cre-ERT)[25]. Tamoxifen-induced cox10 knockout in EndSCLCreERT $cox10^{fl/fl}$ mice ($cox10^{ECKO}$) was verified by PCR (ear biopsies), quantitative real-time (RT)-PCR and Western blotting from freshly isolated ECs (Supplementary Fig. 4a–c). In addition, in order to visualise Cre-mediated recombination, $cox10^{fl/fl}$ mice were additionally intercrossed with a double-fluorescent Cre reporter line R26mTmG. The resulting animals express membrane-targeted tdTomato (mT) prior to Cre-mediated recombination. Cre-mediated recombination induces the expression of membrane-targeted eGFP (mG) in ECs[26]. Upon tamoxifen treatment, endothelial eGFP expression was detected indicating successful recombination in the vasculature of various organs of tamoxifen-treated mice (Fig. 3f). Histological

quantification of blood vessels revealed no difference between $cox10^{ECKO}$ and $cox10^{wt}$ animals (Supplementary Fig. 4d–g). Furthermore, cox10 deletion induced by tamoxifen treatment did not affect survival in adult mice for up to 100 days (Fig. 3g). These data indicate that in the quiescent vasculature of adult mice under homeostatic conditions, endothelial OxPhos is dispensable. Importantly, despite lacking an obvious pathological phenotype, we were unable to culture ECs isolated from adult $cox10^{ECKO}$ animals. We assume that this may be related to the incapacity of cox10-deficient ECs to proliferate in culture as already shown in Supplementary Fig. 3g. To study the angiogenic capacity of ECs isolated from $cox10^{ECKO}$ versus $cox10^{wt}$ mice, we performed the quantitative three-dimensional ex vivo mouse aortic ring assay[27]. Aortic rings isolated from $cox10^{ECKO}$ mice revealed a significant reduction in sprout formation and vascular branching (Fig. 3h–j), indicating that endothelial OxPhos is required for the formation of new blood vessels. Notably, a previous report[3] indicated that the inhibition of mitochondrial OxPhos by using oligomycin in HUVECs does not interfere with the sprouting capacity of ECs, which is in contrast to our observation. However, a follow-up study came to the conclusion, that mitochondrial defects did contribute to EC proliferation and survival[28]. The discrepancy was explained by the use of different experimental setups and timing[28]. Thus, results obtained with chemical inhibitors in cell culture should be considered with special caution[11].

**Neoangiogenesis under disease condition requires EC OxPhos.** Quiescent ECs in the healthy adult are estimated to have a turnover rate of several months[29] whereas ECs involved in neoangiogenesis during wound healing or tumour angiogenesis drastically increase their proliferation rate[30–32]. To explore the role of OxPhos dysfunction in a physiological regenerative process, we first examined the role of EC-specific cox10 deletion in a model of excisional skin wound healing. In this model, timely wound closure depends critically on efficient vascularisation of newly forming granulation tissue. Full thickness excisional punch wounds were introduced on the back of $cox10^{fl/fl}$ and $cox10^{ECKO}$ mice and wound tissue was analysed at 7 days post injury in the phase of tissue formation and proliferation. In order to visualise Cre recombination in this disease model, mice were crossed to the above mentioned Cre reporter line R26mTmG. EGFP expression in ECs of wounds of these mice revealed efficient Cre recombination in $cox10^{ECKO}$ R26mTmG mice while eGFP expression was absent in $cox10^{fl/fl}$ R26mTmG mice (Supplementary Fig. 5a, b). Furthermore, COX/SDH staining demonstrated the functional loss of COX in wound ECs (Supplementary Fig. 5c). Wound closure was delayed in EC-specific cox10-deficient mice compared with floxed control animals as assessed by macroscopic and histological analysis (Fig. 4a–d). While wounds of $cox10^{fl/fl}$ mice

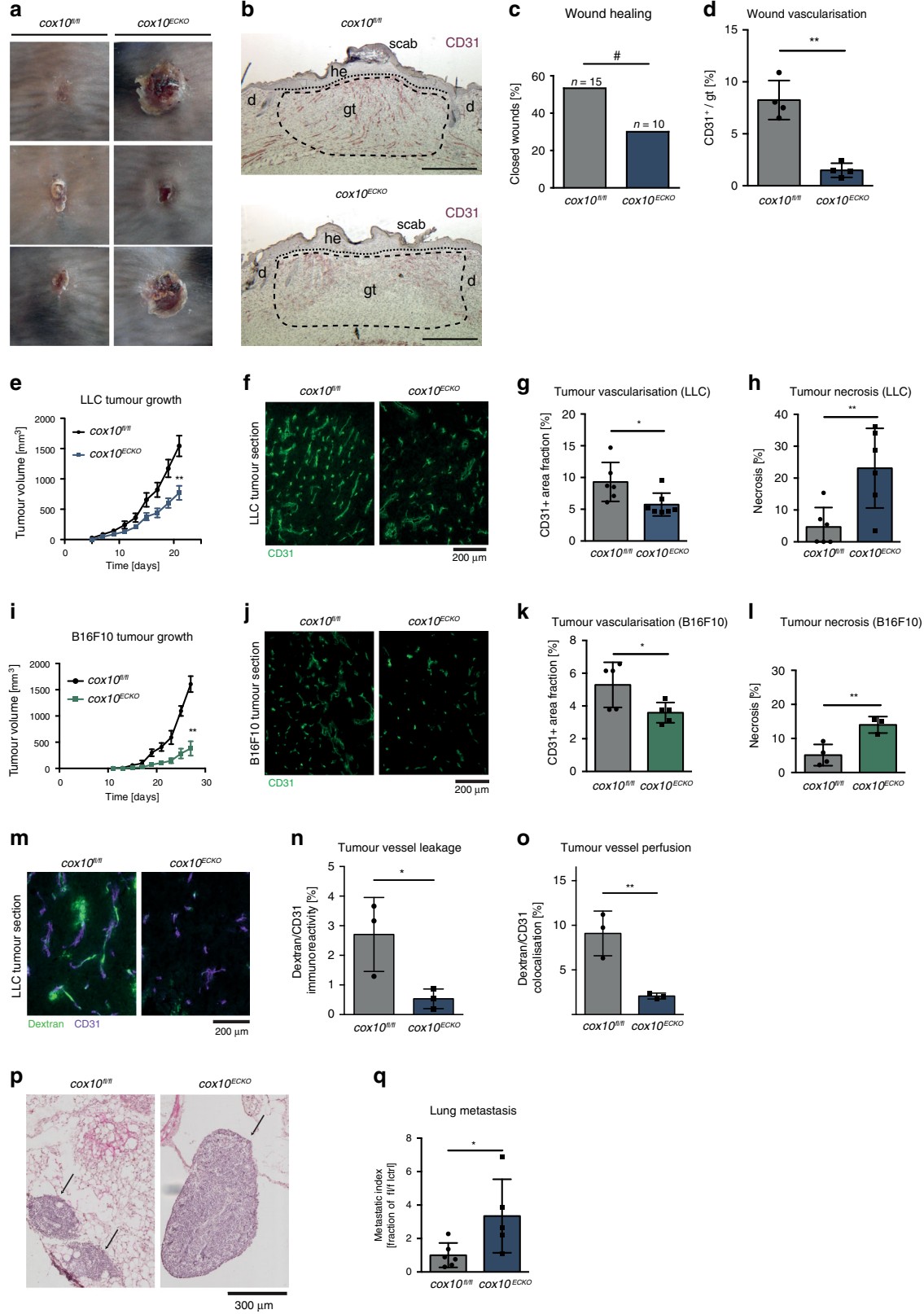

were characterised by a highly vascularised granulation tissue, the vascularisation of granulation tissue in wounds of EC-specific *cox10*-deficient mice was severely impaired (Fig. 4d and Supplementary Fig. 5d). Specifically, a reduction in the vascular density was observed without affecting the pericyte coverage of existing vessels while no pericytes were detectable in the avascular wounds

(Supplementary Fig. 5d, e). These data clearly demonstrate that endothelial OxPhos is required for neovascularisation during wound healing in adult animals.

Next, we examined the role of EC OxPhos in tumour growth and vascularisation in mice. Murine Lewis Lung Cell Carcinoma (LLC) or B16F10 melanoma cells were subcutaneously (s.c.)

**Fig. 4 EC OxPhos is required for neoangiogenesis in wound healing and tumour growth. a** Representative macroscopic wound images of three different mice per genotype (tamoxifen-treated *cox10fl/fl* and *EndSCLCreERT/cox10fl/fl* (*cox10ECKO*) mice). **b** CD31 and haematoxylin stained wound sections derived from these mice at day 7 post injury. d: dermis; gt: granulation tissue; he: hyperproliferative epithelium (scale bar 500 μm). **c** Proportion of closed wounds 7 days post injury. **d** Quantification of CD31+ vessels within the granulation tissue. **e** Lewis lung cell carcinoma (LLC) subcutaneous tumour growth (mean ± SEM) in tamoxifen-treated *cox10fl/fl* mice and *EndSCLCreERT/cox10fl/fl* (*cox10ECKO*) mice. **f** Representative fluorescent images of CD31+ tumour vessels in LLC tumours. **g** Quantification thereof. **h** Quantification of tumour necrosis in LLC tumours. **i** Melanoma (B16F10) subcutaneous tumour growth (mean ± SEM). **j** Representative fluorescent images of CD31+ vessels in B16F10 tumours. **k** Quantification thereof. **l** Quantification of B16F10 tumour necrosis. **m** Representative fluorescent images of CD31+ vessels in dextran perfused LLC tumours. **n** Quantification of dextran leakage and **o** vessel perfusion, **p** Representative H&E-stained images of metastatic lungs from LLC-injected (s.c.) mice. **q** Quantification of the metastatic index. Data are presented as mean ± SD except when indicated otherwise above. Individual data points in (**d**), (**g**), (**h**), (**k**), (**l**), (**n**), (**o**) and (**q**) represent mean values of individual mice of the respective genotype. Sample sizes: **d** $n = 4$, **e** $n = 9$ (*cox10fl/fl*) vs. $n = 7$ (*cox10ECKO*), **g** $n = 6$ vs. $n = 7$, **h** $n = 6$ vs. $n = 6$, **i** $n = 7$ (*cox10fl/fl*) vs. $n = 6$ (*cox10ECKO*), **k** $n = 5$ vs. $n = 5$, **l** $n = 4$ vs. $n = 3$, **n** $n = 3$ vs. $n = 3$, **o** $n = 3$ vs. $n = 3$, **q** $n = 6$ vs. $n = 5$. Exact *p*-value (one sample *t*-test, two-tailed): **c** 0.0424. Exact *p*-values (unpaired students *t*-test, two-tailed): **d** 0.0004, **g** 0.0248, **h** 0.0087, **k** 0.0363, **l** 0.0096, **n** 0.0435, **o** 0.0086, **q** 0.0351. Exact *p*-values (two-way ANOVA, Bonferroni post hoc): **e** <0.0001 (day 21), **i** <0.0001 (day 27).

injected into *cox10ECKO* and *cox10fl/fl* mice following tamoxifen treatment. In both models, tumour growth was significantly reduced in the *cox10ECKO* mice compared with *cox10*-competent hosts (*cox10fl/fl*) (Fig. 4e, i). Both tumour models showed reduced tumour vascularisation while vascular support structures (pericyte coverage) where unaffected (Fig. 4f, g, j, k and Supplementary Fig. 5f, g) as well as increased areas of necrosis (Fig. 4h, l and Supplementary Fig. 5h, i) without significant changes in intratumoural hypoxia (Supplementary Fig. 5j, k). To explore effects on vessel perfusion and leakiness we injected LLC tumour bearing mice with FITC-labelled dextran. *Cox10ECKO* mice showed a significant reduction in tumour vessel leakiness and perfusion (Fig. 4m–o). Tumour vessel disintegration has been shown to promote metastatic progression of tumours[33,34]. Accordingly, we also analysed lungs harvested from LLC s.c. tumour bearing mice for intrapulmonary metastasis and found a significant increase in metastatic index in *cox10ECKO* mice (Fig. 4p, q). Collectively, these data indicated that tumour ECs are dependent on functional OxPhos in order to form tumour vasculature, to supply nutrients and to support tumour growth and vascularisation.

**Metabolic symbiosis between tumour cells and endothelium**. In the tumour microenvironment glucose availability is limited due to the consumption of glucose by highly glycolytic tumour cells. As a result of glycolysis in the hypoxic tumour cells, pyruvate is formed and converted into lactate. It has been demonstrated that this may be utilised by oxygenated tumour cells to fuel OxPhos as a critical regulator of cancer development[35–37]. In line with this concept, we found that ECAR of LLC and B16F10 cells was two to three times higher than that of murine ECs and that these cells produced significant amounts of lactate (Fig. 5a, b). We hypothesised that lactate could also be metabolised by ECs in the tumour microenvironment, in a form of metabolic symbiosis with the tumour cells when glucose availability drops. Indeed, ECs increased their oxygen consumption and the proportion of ATP derived from OxPhos increased in the presence of lactate (Fig. 5c, d). These findings indicate that tumour ECs are capable of metabolising lactate to fuel OxPhos. Similarly, in the spheroid sprouting assay and ex vivo mouse aortic ring assay, *cox10*-competent ECs (*cox10fl/fl*) and aortic rings derived from wild-type animals (*cox10wt*) but not *cox10*-deficient ECs (*cox10KO*) and ECs from aortic rings (of *cox10ECKO* mice) were capable of utilising lactate for energy production and sprout formation (Fig. 5e–h). These observations highlight the propelling role of tumour-derived lactate for EC OxPhos-dependent neoangiogenesis.

## Discussion

Conflicting observations have been made on the relative role of glycolysis versus mitochondrial respiration in ECs[3,8]. Although it is increasingly being recognised that OxPhos is a player in EC metabolism, its significance has been debated[38]. This study seeks to directly address this controversy by genetically disrupting mitochondrial respiratory chain function in ECs in order to assess the relevance of OxPhos for EC function. Our work now provides clear evidence that loss of OxPhos in ECs results in vascular dysfunction leading to (i) embryonic lethality, (ii) impairment of wound vascularisation with delayed wound healing and (iii) a reduction in tumour vascularisation and growth. Our findings support previous studies showing that angiogenic ECs require both, an increase in glycolysis and OxPhos for the full EC angiogenic response[11,37]. Clearly the issue is complicated by the fact that different types of ECs cooperate during angiogenesis, each following a different metabolic programme with different energetic demands. In this respect, two energetically different conditions may be considered; on the one hand, vascular maintenance with low endothelial energy consumption, and full abundance of metabolic substrates—on the other hand, EC sprouting with high energy demand during tumour growth occurring under challenging nutrient conditions.

In the low energy demand situation of vascular maintenance, ECs meet their energetic requirements largely by glycolysis. Although less energy efficient than OxPhos, this has the advantage of reducing the production of potentially damaging reactive oxygen species. In contrast, in the high energy demand situation, e.g. in the tumour microenvironment, where cancer cells and ECs compete for glucose, ATP production by OxPhos becomes increasingly important in ECs.

For cancer cells a symbiotic metabolic interaction has been described whereby cancer cells in hypoxic tumour areas metabolise glucose through anaerobic glycolysis whereas well-oxygenated cancer cells in the vicinity of blood vessels, consume lactate discarded by the hypoxic cancer cells to fuel mitochondrial metabolism[35,39]. Similar symbiotic metabolic relationships have also been observed in the brain, where under homeostatic conditions, lactate derived from glycolytic astrocytes serves as a metabolite for OxPhos in neurons[40,41]. In analogy, our data demonstrate that in the tumour vasculature equivalent interactions appear to operate. Specifically, lactate derived from tumour cells can be metabolised by ECs of the tumour vasculature through OxPhos in order to cover the energetic demands under conditions of limited glucose availability. Together our data show a much greater energetic dependency of the tumour vasculature on OxPhos than has previously been appreciated[38].

During the preparation of this manuscript, Diebold et al. showed that similar to our observations, the conditional knockout

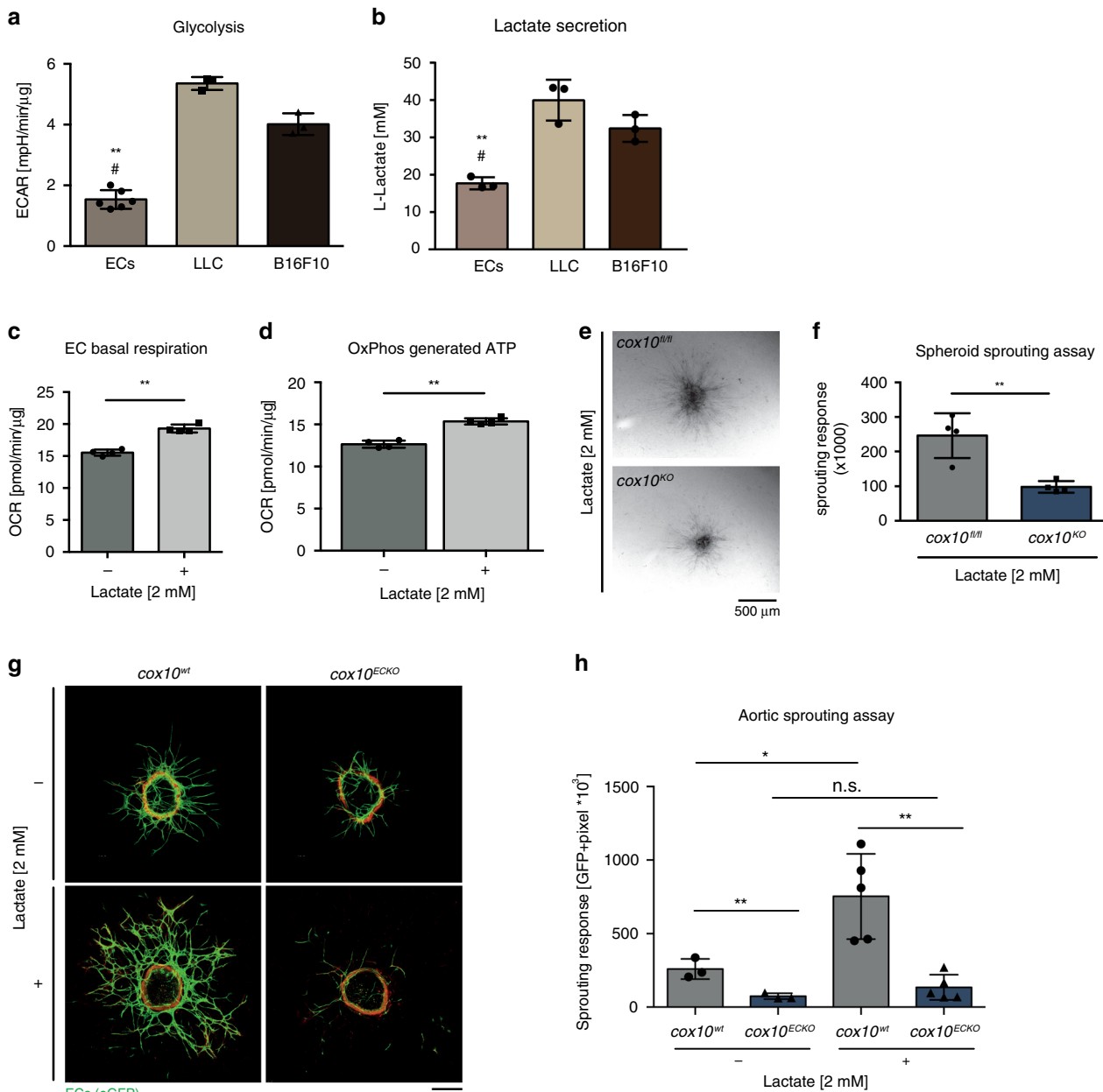

**Fig. 5 Lactate fuels EC respiration. a** Calculated glycolysis rate of murine lung ECs, murine Lewis Lung Cell Carcinoma (LLC) cells and murine B16F10 melanoma cells. **b** Analysis of L-lactate medium levels secreted by ECs and murine tumour cell lines B16F10 and LLC. **c** Basal respiration of isolated murine ECs with or without lactate supplementation. **d** Calculated OxPhos coupled ATP levels of primary murine lung ECs in the absence or presence of lactate. **e** Representative bright field images of sprouting spheroids exposed to lactate. **f** Quantification of sprouting response. **g** Representative aortic ring explants in the presence and absence of lactate. **h** Quantification of the sprouting response. Data are presented as mean ± SD. Individual data points in (**a**), (**b**), (**c**), (**d**) and (**f**) represent technical replicates within a representative experiment. Individual data points in (**h**) represent mean values of individual mice of the respective genotype. Sample sizes: **a** LLC, B16: $n = 3$, ECs $n = 6$, **b** $n = 3$ for all cells, **c** $n = 4$ vs. 4, **d** $n = 4$ for both conditions, **f** $n = 4$ vs. 4, **h** $n = 3$ vs. 3 (−lactate); $n = 5$ vs. 5 (+lactate). Exact $p$-values (unpaired students $t$-test, two-tailed): **c** <0.0001, **d** <0.0001, **f** 0.0044, **h** *=0.0305, **=0.0107, n.s. = 0.2864, **=0.0018. Exact $p$-values (unpaired students $t$-test, two-tailed): ECs compared with both tumour cells (**: ECs vs LLC, #: ECs vs B16F10): **a** **,#: <0.0001, **b** **: 0.0025, #: 0.0030.

of a subunit of the mitochondrial respiratory complex III diminished EC proliferation leading to impaired retinal and tumour angiogenesis[11]. While Diebold et al. also show a requirement for mitochondrial respiration in EC function, their study suggests that the primary role of mitochondria in ECs is to serve as biosynthetic organelles and to safeguard EC proliferation. Our data based on the EC specific disruption of COX in both, embryonic development and adult mice highlight the need for mitochondrial OxPhos for ATP production during angiogenic growth. These findings are supported by our previous demonstration that the energetic dependence of tumour blood vessels on OxPhos can be exploited therapeutically with mitochondrial uncoupling agents[8]. More recently, mitochondrial uncouplers have also been reported to inhibit tumour growth directly by antagonizing the anabolic effects of aerobic glycolysis[42]. The simultaneous targeting of both tumour cells and blood vessels by mitochondrial targeting agents could offer an advantage over conventional cancer therapies and should be explored further.

## Methods

**Mice**. All mouse studies were performed with approval by local government authorities (LANUV, NRW, Germany) in accordance with the German animal protection law. Animals were housed in the animal care facility of the University of Cologne under standard pathogen-free conditions with a 12 h light/dark schedule and provided with food and water ad libitum, temperature was between 20 and 24 °C and relative humidity between 45 and 65 rH. Tie2-Cre mice were received from Dirk Wohlleber[14]. EndSCLCreERT mice were obtained from Joachim R. Göthert[25]. Genotyping PCR for *cox10fl/fl* mice was performed as previously described[13]. In brief, the forward primer (TTCTCTAGGACAGACATTGATGTTTG) targeting the genome in front of the loxP was combined with a reverse primer (GCAGCGCCAGCATCTT) binding exon 6 of COX10 and a reverse primer (GCAGCAAAGAGGGCTCAC) binding behind the second loxP site. Due to the specific PCR parameters, a band occurs at 450 bp without Cre-mediated excision of the loxP-sites flanked exon 6 or at 600 bp after successful excision. Animal genotyping primers are summarized in Supplementary Table 1.

**Embryological studies**. *Tie2-Crewt/wt cox10fl/fl* mice were crossed with *Tie2-Cretg/wt cox10wt/wt* mice. For timed matings a single male was paired with a female mouse which was then plug checked daily. For embryonic staging the day with a positive plug check was counted as E0.5. For whole-mount yolk sac staining, yolk sacs were harvested, washed in PBS, fixed in 4% PFA at 4 °C, washed in PBS + 0.5% Tween-20 (PBS-T), blocked with PBS-T plus 0.2% BSA and 2% normal goat serum (Thermo Fisher Scientific) for 2 h. Yolk sacs were then incubated overnight with directly-labelled anti-CD31 antibodies (Alexa Fluor 647 or FITC anti-CD31, Biolegend MEC 13.3, Supplementary Table 2). Yolk sacs were washed, mounted with Mowiol for imaging (motorised inverted Olympus IX81 microscope (CellR Imaging Software)). Quantification of CD31 was carried out with ImageJ (http://imagej.nih.gov/ij). For whole embryo staining the previously described iDISCO method was employed in combination with Ethyl cinnamate (ECi) tissue clearing[43,44]. In brief, embryos were fixed overnight shaking in 4% PFA at 4 °C. Subsequently, samples were dehydrated in methanol and bleached in 6% H₂O₂ in methanol overnight at 4 °C. Samples were rehydrated in PBS and blocked over 4 days in PBS + 0.2% gelatine + 0.5% TritonX100 (PBSG-T) at room temperature. Samples were incubated in primary rabbit anti-CD31 antibody (1:250, Abcam) for 7 days, followed by intensive washing with PBSG-T and 2-day incubation in secondary Alexa Fluor 647 goat anti-rabbit antibody (1:500, Life Technologies). Tissue clearing was achieved by dehydration in ethanol and incubation in ECi until samples were translucent and subsequently imaged by confocal microscopy.

**Inducible Cre system**. To induce EC-specific Cre expression in *EndSCLCreERT* transgenic mice, tamoxifen was administered by oral gavage on 5 consecutive days 1×/day at a dose of 0.2 mg/g mouse body weight; *cox10fl/fl* or *EndSCLCreERT cox10wt/wt* mice were used as controls. Mice used for experiments were between 8 and 16 weeks old. In tumour or wound-healing experiments, the tumour injection or wounding was performed at day 3 after the last tamoxifen dose.

**Tumour models**. Subcutaneous tumours were generated by injecting $2 \times 10^5$ cells per 100 µl (B16F10) or $1 \times 10^6$ cells per 100 µl (LLC), respectively, into the right flank of recipient mice. Tumour growth was measured every other day for 27 days (B16F10) or 21 days (LLC), starting when tumours reached a size of at least 70 mm³. Tumour volume was calculated as length × width² × π/6[45].

**Cutaneous wound-healing experiments**. Wound-healing experiments were performed as previously described[8,46]. In brief, two punch-biopsy wounds were induced on the shaved back of *cox10fl/fl* or *cox10ECKO* mice under ketamine/xylazine anaesthesia (Ketanest 100 mg/kg body weight, Park Davies; xylazine 2%, 1 ml per kg body weight; Bayer). Tamoxifen was given per oral gavage 5 days in a row, once daily until 3 days before wounding. Wounding and preparation of wound tissue for histology was performed as described previously[46].

**Tumour cells**. LLC and B16F10 melanoma cells were purchased from ATCC. Cells were tested negative for Mycoplasma and other murine pathogens before injection. Cells were maintained in DMEM supplemented with 10% FCS and penicillin/streptomycin (100 U/ml Pen, 100 µg/ml Strep).

**Endothelial cells**. Murine ECs were isolated from lungs of *cox10fl/fl* mice[23]. Organs were resected and briefly washed in PBS, before being minced and enzymatically (0.5% collagenase) digested. Cell solution was then squeezed through a cell strainer (70 µm) and processed for magnetic bead separation (murine CD31, Miltenyi biotec) according to manufacturer's protocol. CD31+ ECs were seeded on gelatin-coated wells and cultured in a 1:1 mixture of EGM2 (PromoCell) and fully supplemented DMEM (containing 20% FCS, 4 g/l glucose, 2 mM glutamine, 1% penicillin/streptomycin (100 U/ml Pen, 100 µg/ml Strep), sodium pyruvate 1% (1 mM), HEPES (20 mM) and 1% non-essential amino acids. After first passage, cells were resorted using the same magnetic beads. EC purification was confirmed by FACS verification, functional assays (sprouting spheroid, tube formation) and RT-PCR of EC-specific and non-specific markers (Endoglin, Acta2), compare also

Supplementary Fig. 2a–c. DNA fragments of *cox10fl/fl* alleles were excised by treatment with recombinant HTNCre for 16 h in a mixture of DMEM/PBS 1:1 (v/v). Complete knockout was confirmed by PCR and western blot analysis. Untreated *cox10fl/fl* cells were used as controls. HUVECs were purchased from PromoCell and cultured in EGM2 (PromoCell). Cells were used for experiments between passages 3 and 8. During experiments ECs were exposed to different levels of glucose concentrations which were either 22, 11, 5.5 or 0 mM D-glucose. When glucose concentration was indicated as 0 it is noteworthy that supplemented serum could theoretically contain very low amounts of glucose (<0.001 mM).

**Metabolic analysis**. To characterise the metabolic changes of isolated ECs, 20,000 cells/well were seeded on gelatin-coated Seahorse 96-well plates 36 h before measurement in a 1:1 mixture of EGM2 (PromoCell) and fully supplemented DMEM and analysed according to Agilent protocols (MitoStress, GlycoStress Kit) with a Seahorse XF96 analyser. Parameters of metabolic function depicted as bar charts in the figures were calculated according to Agilent protocols. Basal respiration: (last rate measurement before oligomycin injection) − ((minimum rate measurement after rotenone/antimycin A injection) = non-mitochondrial oxygen consumption); proton leak: (minimum rate measurement after oligomycin injection) − (non-mitochondrial respiration); ATP production: (last rate measurement before oligomycin injection) − minimum rate measurement after oligomycin injection); spare respiratory capacity: ((maximum rate measurement after FCCP injection − (non-mitochondrial respiration) = maximal respiration) − (basal respiration); glycolysis: (maximum rate measurement before oligomycin injection − last rate measurement before glucose injection); glycolytic reserve: ((maximum rate measurement after oligomycin injection) − last rate measurement before glucose injection = glycolytic capacity) − glycolysis. ECAR and OCR drop was calculated as the difference in the respective parameter before and after addition of glucose in the glycolysis stress assay. To measure oxygen consumption rate with or without lactate the MitoStress was conducted using a modified assay medium without pyruvate and without glucose. Values were normalised to protein concentrations/well determined by Pierce BCA Protein Assay (Thermo Fisher Scientific) according to the manufacturer's protocol and expressed as per µg protein. All Seahorse experiments were performed in the respective Agilent assay medium (assay duration according to manufacturer's protocol up to 90 min with preincubation in a non-CO₂ incubator in non-buffered medium for 1 h).

**L-lactate assay**. In total, $2 \times 10^6$ cells were seeded on gelatin-coated 6-wells in full culture medium. Medium was collected after 24 and 48 h and secreted L-lactate analysed. L-lactate levels were measured using the L-lactate Assay Kit (Cayman Chemical, USA) according to the manufacturer's instructions. Briefly, extracellular lactate concentration was measured by using lactate dehydrogenase for oxidation of lactate to pyruvate, along with the concomitant reduction of NAD+ to NADH. NADH reacts with the fluorescent substrate to yield a highly fluorescent product which was measured subsequently in a plate reader.

**Migration**. Scratch wound assays were performed to evaluate migration on gelatin-coated 96-well plates (20,000 cells/well) in the presence of decreasing glucose concentrations (5.5, 11 and 22 mM). EC monolayers were scratched with a 200 µl tip, stimulated with murine VEGF (Peprotech, 20 ng/ml) and wound closure was analysed at 16 h after scratching. Mitomycin C (2 µg/ml, Cayman) was pre-incubated for 2 h and removed before scratching[11].

**Cell death assay**. In total, 20,000 cells were seeded in endothelial culture medium on 96-wells pre-coated with 1% gelatin. ECs were incubated 24 h in medium with indicated glucose concentrations and treatments in the presence of 0.5 µM yoyo-1 (Thermo Fisher Scientific). Analysis of cell death was carried out using an IncuCyte system (Essen bioscience). Pictures were acquired every 2 h over 24 h with ×10 magnification. Cell death was determined as yoyo-1 positive cells and quantified with the software supplied by the manufacturer.

**Viability assays**. Viability was determined using the neutral red assay[47]. In brief, 20,000 cells/well were seeded on gelatin-coated 96-well plates and cultured in medium overnight. Cells were then exposed to the specific conditions and neutral red assay was performed after 24 or 48 h of incubation.

**HUVEC cell death assay**. In total, 20,000 cells were seeded on gelatin-coated 96-well plates and were treated with oligomycin (2 µM) and antimycin A (10 µM) in the presence (22 mM) and absence (<0.01 mM) of glucose for 15 h. In order to label cell death, cells were imaged with an EVOS FL Auto2 (Life Technologies) using propidium iodide staining solution (50 ng/ml, BD Bioscience) to label dead cells and Hoechst (1.3 µg/ml, Sigma-Aldrich) for identification of the optimal focal plane. Images were acquired every 60 min and quantified utilizing a cell profiler pipeline[48]. Subsequent data visualisation was done with R (http://www.R-project.org).

**Spheroid sprouting assay**. Isolated ECs were cultured as hanging drops in full supplemented DMEM (containing 0.25% methylcellulose) overnight to generate

spheroids (1000 cells/spheroid) which were then embedded in a glucose-containing or glucose-free collagen matrix. Sprouting was induced by addition of murine VEGF (30 ng/ml, Peprotech) containing 22 mM glucose-containing or 11 mM medium. Sprouting response was analysed after 48 h. Sprouting assays in the presence of lactate were conducted with 2000 cells/spheroid and the sprouting response (in full supplemented glucose-free DMEM) was analysed after 72 h.

**Proliferation.** Proliferation of ECs was assessed using the CellTrace CFSE Cell Proliferation Kit, for flow cytometry (Thermo Fisher Scientific) according to the manufacturer's instructions. In total, 40,000 cells were seeded in 12-wells and stained with CellTrace™ CFSE. The fluorescent staining was analysed by fluorescence-activated cell sorting (FACS) after 24 and 72 h using a BD FacsCanto (BD BioSciences). The cellular concentration of this compound is diluted in each cell division. Therefore, fluorescence intensity negatively correlates with the number of cell divisions. The intensity after 24 h was divided by the signal strength at 72 h and plotted as dilution factor.

**Aortic ring assay.** Aortic ring assays were performed as previously described[27]. In brief, the thoracic aorta was excised from tamoxifen-treated $cox10^{fl/fl}$ or wild-type mice that were $EndSCLCreERT R26mTmg$ transgenic. Rings were starved overnight in Opti-MEM (Thermo Fisher scientific, containing 1% penicillin/streptomycin), embedded in rat collagen (final use concentration 1 mg/ml; Millipore, Merck); (1 ring per well/96-well plate) and medium containing 30 ng/ml murine VEGF (Peprotech) was added. For aortic ring sprouting in the presence or absence of lactate fully supplemented DMEM without glucose and without Na-pyruvate was used. Rings were cultured for 5 days. Since the endothelium of these aortic rings expressed eGFP, the sprouting response could be visualised by life fluorescence microscopy without fixation.

**Confocal fluorescence microscopy.** eGFP positive endothelial cells from whole tissue mounts of mouse ear (fixed for 10 min with 4% formalin), aortic rings (unfixed) and immuno-stained whole embryos were imaged using a confocal laser scanning or multiphoton microscope (confocal: TCS SP8, Leica Microsystems; MP: TCS SP8 MP-OPO (Leica Microsystems). A 10× dry objective (HC PL APO 10×/0.40 CS2, Leica Microsystems) with a numerical aperture of 0.4 or a 5× air objective (PL FLUOTAR 5×/0.15, Leica Micosystems) with a numerical aperture of 0.15 was used and the samples were illuminated with solid state laser sources at 488, 552 and 638 nm, respectively. The fluorescence signals were collected using sensitive HyD detectors. To image the complete sample, the tile scan mode of the microscope was used, and z-stacks were acquired with a step size of 5 µm. The acquired z-stacks were finally processed as maximum projections to give overview images. For multiphoton imaging a 25× water immersion objective (IR Apo L25×/0.95W) with a numerical aperture of 0.95 was used and the samples were illuminated using a tuneable MP-laser (Chameleon Vision II, Coheren).

**Immunohistochemistry.** Immunohistochemical analyses were performed on cryosections as previously described[8,49,50]. Briefly, organs or tumours were resected, embedded in O.C.T. compound, Tissue-Tek, and stored at −80 °C. Cryosections were dried, fixed in 4% PFA, washed, blocked with 10% normal goat serum and were incubated with the respective antibodies. To analyse organ vasculature, cryosections (10 µm) from hearts, livers and kidneys were stained for CD31 with a monoclonal Armenian-hamster anti-mouse CD31 antibody (1:200, Abcam) and a secondary Alexa Fluor 647 goat anti-hamster antibody (1:500, Abcam).

For histological characterisation of tumour vascularisation, cryosections (20 µm) were stained for CD31 (monoclonal Armenian-hamster anti-mouse CD31 antibody (1:200, Abcam), secondary Alexa Fluor 647 goat anti-hamster antibody (1:500, Abcam); tumour cell nuclei were stained with 4,6-diamidin-2-phenylindol (DAPI). Pericyte coverage (PDGFRβ, rat anti-PDGF-R antibody (1:100, CD140b, eBioscience)) was analysed on 100× images and quantified with the ImageJ colocalisation plugin[49]. For further reference, all antibodies used in this study are summarized in Supplementary Table 2. Imaging was conducted on a motorised inverted Olympus IX81 microscope (Cell[R] Imaging Software). Area densities were calculated using ImageJ (http://imagej.nih.gov/ij). For assessment of perfusion/leakiness and hypoxia, mice were intravenously injected with FITC-dextran (100 µl/25 g mouse body weight, at a concentration of 15 mg/ml, molecular weight 2000 kDa, Sigma) 30 min before sacrifice and intraperitoneally injected with pimonidazole (100 µl/25 g mouse body weight, at a concentration of 15 mg/ml) 60 min before sacrifice. Tumour vessel leakage was calculated as dextran area density per CD31 area density. Tumour vessel perfusion was calculated using the ImageJ colocalisation plugin to determine dextran that colocalised with CD31 positive. Colocalisation was considered as perfused vessel. Pimonidazole adducts were detected with a FITC-labelled anti-pimonidazole antibody to visualise hypoxia.

Metastatic lungs of LLC-injected mice were H&E-stained and serial sections were generated. Every 10th section was analysed for metastatic area with ImageJ. Metastatic index was calculated as metastatic area per primary tumour volume.

For histological analyses in the wound-healing model, wounds were excised at 7 days after injury and bisected in caudocranial direction. The tissue was either immediately embedded in OCT (Tissue-Tek) or fixed for 2 h in 4%

paraformaldehyde (in PBS) and subsequently embedded in OCT. Wound tissue was stored at −80 °C. Morphometric analysis was performed on H&E-stained tissue sections using light microscopy equipped with a KY-F75U digital camera (JVC) at various magnifications (Leica DM4000B, Leica Microsystems; Diskus 4.50 software)[46,51,52]. Briefly, the extent of granulation tissue formation was determined between the epithelial tips and analysed as a measure of wound closure. For anti-CD31 immunostaining 5-µm cryosections were fixed in acetone and incubated with NaN$_3$ and H$_2$O$_2$ in PBS to block endogenous peroxidase and with 10% FCS in PBS to block unspecific antibody binding. Sections were incubated for 1 h at RT with the primary anti-mouse CD31 antibody (Becton Dickinson), diluted 1:1000 in PBS/1% BSA. Sections were washed with PBS and incubated for 1 h at RT with secondary goat anti-rat-HRP antibody (Life Technologies), diluted 1:250 in PBS/1% BSA. Sections were washed with PBS and AEC substrate was used to detect CD31-bound HRP. Counterstaining was performed with haematoxylin. Analysis of the CD31+ area was carried out using a Leica DM4000B microscope as described above and ImageJ software.

**Western blot.** For whole-cell lysates, collected cell pellets were resuspended in 1 pellet volume with CHAPS lysis buffer (10 mM HEPES pH 7.4, 150 mM NaCl, 1% (w/v) CHAPS, protease inhibitor (complete Mini, Roche)), incubated for 20 min on ice and subsequently centrifuged at $20,000 \times g$ for 20 min at 4 °C. For supernatants —representing whole-cell lysates—protein concentration was determined with Pierce BCA Protein Assay (Thermo Fisher Scientific). For Western blotting, equal amounts of proteins were resolved by SDS-PAGE in a Mini-PROTEAN Electrophoresis System (Bio-Rad) and transferred onto nitrocellulose membrane (Amersham Protean, GE) by a Mini-Trans-Blot System or Trans-Blot Turbo Transfer System (Bio-Rad). Membranes were incubated with the appropriate primary and secondary antibodies. After antibody incubation, membranes were developed using a ChemiDoc MP Imaging System (Bio-Rad)[53].

**Metabolomics analysis.** Levels of nucleotides and intermediates (organic acids, sugar phosphates) of the glycolysis and the citric acid cycle in murine endothelial cells were determined by anion-exchange chromatography coupled to electrospray ionization high-resolution mass spectrometry (IC-ESI-HRMS) using a procedure previously described[54] with several modifications: $1.3 \times 10^6$ cells per biological replicate were suspended in 400 µl of ice-cold methanol/water 4:1 (v/v). Ten microlitres of a mixture of isotope-labelled internal standards in Milli-Q water (50-µM 13C10-adenosine 5′-triphosphate (13C10-ATP), Sigma-Aldrich; 5-µM 13C6-D-glucose-6-phosphate (13C6-G6P) and 5-µM D4-succinic acid (D4-SUC), both Eurisotop) were added. After thorough mixing and centrifugation ($16,100 \times g$, 5 min, 4 °C), 350 µl of supernatant were dried under reduced pressure. The residue was resolved in 100 µl of Milli-Q water, transferred to autoinjector vials and immediately measured.

IC-HRMS analysis was performed using a Dionex Integrion RFIC system (Thermo Scientific) equipped with a Dionex IonPac AS11-HC column (2 mm × 250 mm, 4 µm particle size, Thermo Scientific) and a Dionex IonPac AG11-HC guard column (2 mm × 50 mm, 4 µm, Thermo Scientific) and coupled to a Q Exactive HF quadrupole-orbitrap mass spectrometer (Thermo Scientific). Five microlitres of sample were injected using a Dionex AS-AP at 10 °C. The IC was operated at a flow rate of 0.38 ml/min with a potassium hydroxide gradient which was produced by an eluent generator with a potassium hydroxide cartridge and Milli-Q water. The gradient started with 10 mM KOH over 3 min, 10–50 mM from 3 to 12 min, 50–100 mM from 12 to 19 min, held at 100 mM from 19 to 25 min, and re-equilibrated at 10 mM for 3 min. The total run time was 28 min. A Dionex ADRS 600, 2 mm suppressor was operated with 95 mA, and methanol was used to produce a make-up flow at a flow rate of 0.15 ml/min.

The mass spectrometer was operated in the negative ion mode. Full MS scans in the range of $m/z$ 60–900 were acquired with a resolution of 60,000, an automatic gain control (AGC) target value of $1 \times 10^6$ and a maximum injection time (IT) of 100 ms. Spectrum data were collected in the centroid mode. The ESI source was operated with flow rates for sheath gas, auxiliary gas and sweep gas of 50, 14 and 3, respectively. The spray voltage setting was 2.75 kV, the capillary temperature 230 °C, the S-lens RF level 45, and the auxiliary gas heater temperature 380 °C.

The exact $m/z$ traces of the internal standards and endogenous nucleotides, organic acids, and sugar phosphates were extracted and integrated using the Target Screening workflow within the TraceFinder 4.1 software (Thermo Scientific). Endogenous metabolites were quantified by normalising their peak areas to those of the internal standards: 13C10-ATP was used as internal standard for endogenous nucleotides, 13C6-G6P for sugar phosphates, and D4-SUC for organic acids. The normalised peak areas were related to the mean values of the control cells (relative abundance).

**RNA isolation and quantitative real-time PCR (qRT-PCR).** RNA was isolated with the RNeasy Kit (Qiagen) according to manufacturer's instructions and RT-PCR was performed using the Maxima H Minus First Strand cDNA synthesis Kit (Thermo Scientific). Subsequently quantitative real-time PCR was used to measure expression levels of indicated genes. Samples were measured in technical triplicates in a 96-well plate Multicolour real-time PCR Detection System (IQ™5, Bio-Rad) using LightCycler®SYBR-Green I Mix (Roche). Data analysis was done based on

linear regression of the logarithmic fluorescence values/cycle with the programme LinRegPCR[55,56] and target gene expression was normalised to the reference gene *Actin*. Primers used:

Actin_FW: CTGAGAGGGAAATCGTGCGT
Actin_RV: AACCGCTCGTTGCCAATAGT
Cox10_FW: AAGAACAGGCCTCTGGTTCG
Cox10_RV: GCTCCAACCCAGGTATTGGT
Endoglin_FW: CCCTCTGCCCATTACCCTG
Endoglin_RV: GTAAACGTCACCTCACCCCTT
Acta2_FW: GTCCCAGACATCAGGGAGTAA
Acta2_RV: TCGGATACTTCAGCGTCAGGA

**Statistics and reproducibility**. Data are presented as mean ± SD except stated otherwise in the figure legends. Sample sizes (replicates, animals) are traceable as individual data points in each figure. In vitro experiments were repeated at least two times. Data involving animals depict pooled data of at least three independent experiments. All statistical tests used to examine statistical significance were two-sided, confidence interval: 95% (where applicable). Exact *p*-values and the respective test/analysis are listed in the figure legends. Microsoft Excel was used for data collection and GraphPad Prism 6 and R (http://www.R-project.org) were used to analyse data in this study.

**Reporting summary**. Further information on research design is available in the Nature Research Reporting Summary linked to this article.

## Data availability

The data supporting the findings of this study are available from the corresponding author upon reasonable request. Raw metabolomics data are available on MetaboLights repository project number MTBLS1764 (https://www.ebi.ac.uk/metabolights/MTBLS1764). Source data are provided with this paper.

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

## Acknowledgements

H.K. acknowledges funding from the Deutsche Krebshilfe (70112113) and the Deutsche Forschungsgemeinschaft SFB1218 (project number 269925409) and SFB1403 (project number 414786233). L.M.S and F.H.K. are supported by the Koeln Fortune Programme, Faculty of Medicine, University of Cologne. S.A.E. acknowledges funding from the Deutsche Forschungsgemeinschaft (DFG, German Research Foundation Project ID 73111208-SFB829, FOR2240, FOR2599 and DEBRA International). We thank Joachim Göthert (University Hospital Essen, Germany) and the Telethon Kids Institute (Subiaco, Australia) for providing the EndSCLCreERT transgenic mice and Dirk Wohlleber (Technical University of Munich, Germany) for providing the Tie2-Cre mice. We thank the CECAD Imaging Facility, University of Cologne (Head: Dr. Astrid Schauss). We thank Tanja Roth, Ali Manav, Maureen Menning and Michael Piekarek for great technical assistance.

## Author contributions

Study design: L.M.S., O.C. and H.K.; providing of study materials or samples: L.M.S., M.P., M.K., C.T.M., S.A.E., O.C. and H.K.; data generation, analysis and interpretation: L.M.S., J.P.W., F.H.K., J.M.S., S.W., S.D.G., M.C.A., F.S., M.F., S.B., C.L., C.J.B., M.K., S.A.E., O.C. and H.K.; manuscript writing: all authors.

## Competing interests

The authors declare no competing interests.
