## [Peer Review File · Nature Communications]

Reviewers' comments:

Reviewer #1 (Remarks to the Author):

This is an interesting paper presenting convincing data that support the crucial role of OXPHOS in angiogenesis during embryonic development, or angiogenic stimulation linked to classical models such as wound healing or tumour growth. The results are sound and the paper is clear and well written. I have two points that should be addressed. First, the clinical phenotype of tamoxifen-inducible EC KO should be specified and possibly expanded, with the inclusion of physiological stimuli that could affect angiogenesis, for instance endurance training (treadmill tests) or learning. Second, the angiogenic stimuli are convincing, however the dependency of EC cells to glycolysis rather than OXPHOS could be challenged metabolically, for instance to evaluate the clinical and histological phenotype under a strictly chetogenic-high fat diet in order to force in vivo OXPHOS rather than glycolysis. Minor point: some of the concentrations used in the cell biology section should be explained, for instance the concentration of lactate (2mM) used in some experiments.

Reviewer #2 (Remarks to the Author):

In this study, the authors evaluate the role of mitochondrial respiration in neoangiogenesis during development and pathological conditions. They report that EC-specific *cox10*-deficiency impairs vascular development and causes embryonic lethality in mice, and observe respiratory dysfunction measured in the primary cultured ECs in vitro (Fig 1). Interestingly but not surprisingly, the ECs with *cox10* deletion are more susceptible than WT ECs to glucose deprivation as the KO ECs are unable to compensate the loss of glucose availability (Fig 2). The dramatic phenotype seen during development is absent if *cox10* is deleted in adulthood suggesting that EC Oxphos is dispensable for the maintenance of vascular density under normal healthy condition (Fig 3). During wound healing or tumor angiogenesis, conditions that involve neovascularization, in contrast, animals with *cox10* EC deletion exhibits defect in neoangiogenesis (Fig 4).

Overall, this is a well-executed study (with some caveats below), but largely descriptive with little mechanistic insights. The authors claim in the abstract that this is the first genetic evidence for a requirement of mitochondrial respiration for neoangiogenesis, but a recently published study shows that endothelial respiration is critical for angiogenesis with deletion of a different ETC complex, and demonstrates a mechanistic role for anaplerosis (Diebold et al., Nature Metabolism 2019), and therefore the current study does not break fundamental new ground. Comparison of the current

work with that study (where findings are similar but with some differences) would be of interest.

Additional comments:

1. Validation of the KO mice is lacking and is absolutely necessary in light of some of the negative results: evidence of neither efficiency nor specificity of their KO in neither Tie2Cre nor EndSCLCreERT line is provided. It is critical to prove that the phenotypes that the authors observed in the mice are due to the deletion in ECs and not due to the deletion in other cell types such as bone marrow cells (common with Tie2Cre, for example). In addition, confirming efficient KO would be especially critical if the authors want to claim dispensableness of a gene of interest. No data on validation of Cox10 knockout in ECs are provided (eg. qPCR, WB, IHC analysis of Cox10 in the WT vs KO mice).

2. The migration defect that the authors observed could be attributed by the defect in viability. The authors should provide data that viability is intact at the end of migration experiments or remove the claim on the defect in migration. The viability data that the authors provide in the current manuscript is only for a short period of time (15 hrs) while the migration assay was conducted for a longer period of time (48 hrs).

3. The method section needs to be revised to include significant details. The method on tumor necrosis assay (Fig 4k-l), for example, is not provided in the manuscript.

Reviewer #3 (Remarks to the Author):

General:

Angiogenesis or the formation of new blood vessels from pre-existing ones, is crucial for development and wound healing but is also involved in tumor vascularization. During angiogenesis, endothelial cells (ECs, lining the lumen of blood vessels) start proliferating and migrating while undergoing a metabolic switch, characterized for example by an increased glycolysis rate to fulfill the increased energy demand. The role of mitochondrial oxidative phosphorylation (OxPhos) in angiogenesis remains debated. In this manuscript, Schiffmann et al. describe that EC-specific deletion of *cox10*, causing an EC-restricted OxPhos deficiency, leads to embryonic lethality but does not affect the quiescent vasculature in healthy adult mice. However, in mouse tumor models, EC-specific *cox10* knockout reduces vascularization and tumor growth. Mechanistically, the authors propose that during angiogenesis, proliferative ECs in a glucose-deprived environment require OxPhos for energy production for which they use lactate coming from hyper-glycolytic tumor cells.

While possibly of interest to the broad readership of this journal, the manuscript suffers from a number of shortcomings and overstatements. The following comments will need to be addressed.

General/major comments:

1. A first major concern pertains to the novelty of the presented work. A crucial role for endothelial OxPhos in developmental and pathological angiogenesis has recently been described by Diebold et al. (2019). The authors thus obviously overstate their case when writing upfront in the abstract: "Our data provide the first genetic evidence for a requirement of mitochondrial respiration for neoangiogenesis during development and disease." In mice with an EC-specific knockout of complex III subunit *Uqcrcq*, Diebold et al. showed that OxPhos is required for postnatal retinal and lung vascularization and tumor angiogenesis. They showed the underlying mechanism to be a decreased NAD⁺/NADH ratio which inhibits the TCA cycle flux and thus production of amino acids, and finally impairs EC proliferation but not migration. Did the authors make similar observations for their EC-specific *Cox10* knockout? Evaluating one or two key findings from the Diebold paper in the EC-specific *Cox10* KO setting, would offer the readers a most interesting view on (dis)similarities between deletion of different OxPhos components in ECs.

2. Along the same lines, the authors can further increase novelty of their work by establishing the *cox10*-deficient embryonic phenotype in more detail (this was left unstudied in the Diebold paper). For instance, different parameters of the vasculature can be studied in whole-mount embryos (rather than in yolk sacs), and physiological parameters like heartbeat can be assessed to obtain insights into overall development of the cardiovascular system.

3. The authors claim that OxPhos is required to meet the energy demands of ECs but show no hard evidence for this statement, e.g. by measuring intracellular ATP levels or energy charge in *cox10*^{-/-} ECs. Moreover, they did not formally exclude that the underlying mechanism is a defect in TCA cycle (Diebold et al. 2019) or pyrimidine synthesis.

4. The choice for the high glucose level used in the in vitro assays needs better justification; 22 mM is not simply high but is almost hyperglycemic. The data from the different assays using these high glucose levels would strongly benefit from inclusion of a condition representing more physiologically relevant glucose levels. For the low or zero glucose conditions, was dialyzed serum used in the medium to avoid that remnant levels of glucose coming from the serum are present in the medium and are not accounted for?

5. In their embryological studies, the authors used the *Tie2*-Cre transgenic mice to drive EC-specific Cre expression. As *Tie2*-Cre is not only specific for ECs but also hits the hematopoietic line, it is strongly advisable to confirm key findings with a more specific Cre driver line (e.g. the *EndSLC*-CreERT model used for the adult mouse studies or other well established EC-specific Cre drivers, e.g.: *Flk*:Cre (PMID: 14745955) or *VE-Cadh*:Cre (PMID: 16450386). At the very least, the limitation of using the *Tie2*-Cre should be clearly acknowledged.

6. According to the main text, no obvious phenotypic defects in *cox10*-deficient ECs were observed in the in vitro studies. The authors should show supporting data, investigating for instance EC apoptosis, proliferation, senescence or oxidative stress. This also contradicts the authors' statement on the 'incapacity of *cox10*-deficient ECs to proliferate in culture', which is not in line with the observation that OxPhos is only crucial under glucose-limitation. This is all rather confusing, raises questions about experimental conditions and needs further explanation.

7. Fig. 2f-i: The authors showed that low glucose conditions significantly reduce the viability of *cox10* deficient ECs, yet they use these low glucose conditions in scratch wound migration and spheroid assay. How can the authors make sure that the alleged migration defect observed in these assays is not simply due to reduced viability/proliferation of the *cox10* deficient ECs? Also, additional assays specifically assessing EC proliferation or migration (e.g. Boyden chamber) should be performed to be able to make a more sound conclusion on this matter. This is especially relevant, since this statement contrasts with observations made by Diebold et al. who noted a proliferation but no migration defect in OxPhos-deficient ECs.

8. The paper would largely benefit from additional analyses of blood vessel parameters in the models for physiological and pathological angiogenesis, e.g. metastasis index for tumor models, hypoxia, vessel maturation (pericyte coverage), vessel perfusion/ leakage (Dextran) or cell proliferation (e.g. Ki67).

9. Fig. 4o-q: These figure panels make little sense without inclusion of a control condition (0 mM lactate added) for both genotypes.

Minor comments:

1. From the methods section, it is clear that the OCR and ECAR data were normalized to total protein content (BCA). It is then surprising to see that the data in the graphs are expressed as normalized to arbitrary units (a.u.). Can the authors express the OCR and ECAR data per microgram or milligram protein – which is a much more widely accepted way of presenting these data and which allows for comparison of OCR and ECAR values between independent studies.

2. Fig. 1c: The authors should present quantifications for the number of blind ends and irregular vessel dilatations in the yolk sacs, rather than simply stating in the main text that the occurrence of these two features was increased in the KOs. The reader might also benefit from an explanation of the yellow and red arrows in the figure legend.

3. Fig. 1e-i: The authors should present data on the purity of EC populations isolated from *COX10^{fl/fl}* mice which were used for the in vitro studies.

4. Ext. Fig. 1c: The authors claim that also the vasculature in E10.5 yolk sacs presents severe malformations. However, these are not at all obvious in the representative images. Therefore, quantifications (vascular density, number of blind ends, number of dilatations...) will need to be added to these figure panels in order to make a convincing case here.
5. Ext. Fig. 1e: The exact same data (except for the heterozygous condition) is presented in Fig. 1d. To avoid all possible confusion, it might be best to explicitly acknowledge this in the legend.
6. Fig. 2: The method section lacks a clear description of the induction of endothelial quiescence, only a reference is mentioned.
7. Fig. 2a: Please state to which control condition the drop/gain of OCR and ECAR is compared.
8. Fig. 2a: The authors state in the main text "Thus, under saturating amounts of glucose, OxPhos in COX10-competent ECs was suppressed and they metabolically behaved like COX10-deficient ECs." This should be confirmed by presenting the OCR and ECAR measurements of COX10 competent and deficient ECs exposed to the different glucose concentrations in the same experiment, side by side.
9. Fig. 2b,d,e,f,g,h,i: Besides analyzing ECs exposed to high and low glucose concentrations, the assays should also be performed in normal glucose conditions (see main comment above).
10. Fig. 2f: Did the authors use low glucose conditions (as noted in the figure) or no glucose (as noted in material & methods)? Typographical errors are present in the y-axis legend (the same holds true for Fig. 2h).
11. Fig. 3: Efficient recombination of the loxed alleles will also need to be shown for the EndSCL-CreERT driver.
12. Fig. 3f: Please adjust the axis title ("junctions" versus "vascular nodal points" in legend).
13. Ext. Fig. 3: Quantifications of the vascular area in all presented organs will need to be added. The authors should consider using colored images for clarity.
14. Fig. 4c: To get a better understanding of possible differences in the dynamics of wound healing between genotypes, the authors could assess additional parameters besides the total percentage of animals with closed wounds, like for instance evolution of the wound diameter over time.
15. Fig. 4h: Statistical analysis needs to be added – the tumor volumes between the two genotypes seem clearly different, yet no statistical analysis is provided.
16. Fig. 4m: OCR data of cox10-deficient ECs with or without lactate supplementation will need to be added.
17. Fig. 4o,q: Please use uniform output parameters for sprouting and aortic ring analysis (e.g. sprout number/length, extra-aortal GFP+ area; in the same way as presented in figures 2 and 3).
18. Ext. Fig. 4a,b: Please enlarge the pictures as well as the asterisks, for clarity.
19. Ext. Fig. 4c: This analysis would benefit from measuring the actual lactate levels in the medium of the different cell types.

20. Legends: Please include the sample size.

Point-By-Point Response NCOMMS-19-01883-T

Reviewer #1 (Remarks to the Author):

This is an interesting paper presenting convincing data that support the crucial role of OXPHOS in angiogenesis during embryonic development, or angiogenetic stimulation linked to classical models such as wound healing or tumour growth. The results are sound and the paper is clear and well written. I have two points that should be addressed. First, the clinical phenotype of tamoxiphen-inducible EC KO should be specified and possibly expanded, with the inclusion of physiological stimuli that could affect angiogenesis, for instance endurance training (treadmill tests) or learning. Second, the angiogenetic stimuli are convincing, however the dependancy of EC cells to glycolysis rather than OXPHOS could be challenged metabolically, for instance to evaluate the clinical and histological phenotype under a strictly chetogenic-high fat diet in order to force in vivo OXPHOS rather than glycolysis. Minor point: some of the concentrations used in the cell biology section should be explained, for instance the concentration of lactate (2mM) used ind some experiments.

We thank this reviewer for the evaluation of our work and intriguing suggestions. We are indeed very interested in further exploring the role of OxPhos in ECs under physiological conditions such as endurance training or specific-dietary restrictions as suggested. To this end, we have been able to obtain some of the necessary licence amendments to address the reviewer's suggestions and have already initiated ketogenic-high fat diet experiments. The first set of these analyses was just completed (see Figure 1 Reviewer #1 below). Unfortunately, the inclusion of endurance training (treadmill tests) was not possible in a time frame compatible with this revision.

As requested we performed ketogenic-high fat diet in line with previously described experimental setups^{1,2} and exposed *cox10* competent and EC specific *cox10* deficient mice to control diet and experimental diet. We analysed these mice regarding clinical and pathological parameters (see below). These analyses however did not identify gross phenotypic alteration when OxPhos was blocked in ECs. We could only detect slightly reduced glucose levels in blood of *cox10* deficient mice subjected to an experimental diet (c). Notably, some wild type as well as COX-deficient mice, did not feed adequately on ketogenic diet and lost significant weight during the course of the experiment and could thus not be included in our current analysis (we initially included 15 mice for each group). We apologise that at this stage, we were therefore unable to formally confirm the metabolic switch in these mice and considering their preliminary nature, we did not yet include these results in our revised manuscript. We estimate that the execution of the requested animal experiments will require an additional 6-8 months for the breeding and feeding strategy. In light of this, we would kindly ask the reviewer to reconsider the importance of these additional mouse experiments.

Physiological blood lactate concentration normally is maintained below 2 mmol/L. We used the concentration of 2 mmol/L to represent the threshold between the physiological blood lactate levels and the pathological concentrations that ECs are exposed to in the tumour microenvironment, which was confirmed by our own new measurements in the supernatants of tumour cells *in vitro* (new Fig. 5b).

Figure 1. Clinical parameters of $cox10^{fl/fl}$ and $cox10^{ECKO}$ animals subjected to high fat diet. **a**, Kaplan-Meier survival curve of $cox10^{fl/fl}$ and $cox10^{ECKO}$ animals fed with control or high fat diet after 17 days of treatment. **b**, weight change [%] of $cox10^{fl/fl}$ and $cox10^{ECKO}$ animals fed with control or high fat diet. **c**, lactate and glucose concentrations in the blood of animals with indicated genotypes fed with either control or high fat diet. **d**, representative 10x images of blood vessel (CD31) in heart, liver or kidney of animals, fed with control or high fat diet (upper panels). The respective quantification of blood vessel density (lower panels). **e**, representative images of 40-fold magnified blood vessels in kidneys of $cox10^{fl/fl}$ and $cox10^{ECKO}$ animals fed with control diet. **f**, PCR-analysis of $cox10$ alleles in $cox10^{fl/fl}$ and $cox10^{ECKO}$ mice. Data are presented as mean \pm SD. Individual data points in **(b)** and **(c)** represent values of individual mice of the respective genotype and treatment group. Individual data points in **(d)** represent mean values of individual mice of the respective

genotype. Sample sizes: (b), (c) from left to right: n=7, n=8, n=4, n=7. (d) n=3. Log rank test was used to determine differences between survival curves in (a), exact p-value: 0.5418. Exact p-values (unpaired students t-test, two tailed): (b) 0.6919, 0.4682 (c) 0.4540, 0.8597, 0.9068, 0.027.

Reviewer #2 (Remarks to the Author):

In this study, the authors evaluate the role of mitochondrial respiration in neoangiogenesis during development and pathological conditions. They report that EC-specific *cox10*-deficiency impairs vascular development and causes embryonic lethality in mice, and observe respiratory dysfunction measured in the primary cultured ECs in vitro (Fig 1). Interestingly but not surprisingly, the ECs with *cox10* deletion are more susceptible than WT ECs to glucose deprivation as the KO ECs are unable to compensate the loss of glucose availability (Fig 2). The dramatic phenotype seen during development is absent if *cox10* is deleted in adulthood suggesting that EC Oxphos is dispensable for the maintenance of vascular density under normal healthy condition (Fig 3). During wound healing or tumor angiogenesis, conditions that involve neovascularization, in contrast, animals with *cox10* EC deletion exhibits defect in neoangiogenesis (Fig 4).

Overall, this is a well-executed study (with some caveats below), but largely descriptive with little mechanistic insights. The authors claim in the abstract that this is the first genetic evidence for a requirement of mitochondrial respiration for neoangiogenesis, but a recently published study shows that endothelial respiration is critical for angiogenesis with deletion of a different ETC complex, and demonstrates a mechanistic role for anaplerosis (Diebold et al., Nature Metabolism 2019), and therefore the current study does not break fundamental new ground. Comparison of the current work with that study (where findings are similar but with some differences) would be of interest.

We thank this reviewer for his/her constructive criticism. The reviewer correctly points out, that during the revision period of our manuscript which was originally submitted in 2018 (*Nat Cell Biol.*) but was redirected for review first to *Nat. Metabolism* and then further to *Nat. Commun.* - the work by Diebold *et al*³ was accepted for publication and published earlier in 2019. In our revised manuscript we acknowledge the important study by Diebold *et al*, and discuss the potential impact of our findings with reference to this work.

We specifically highlight converging and diverging conclusions made in the two independent studies that superficially appear to employ similar experimental designs, but substantially differ in some results and interpretation of the findings. In contrast to our work, the study of Diebold *et al* suggested that functional aspects of ECs such as sprouting and tube formation were *not* affected by inhibition of OxPhos, based on results obtained with HUVECs treated with antimycin A in a spheroid assay. Addressing this issue, we now include new metabolic characterizations of ECs (Fig. 2j) lacking COX and provide additional evidence that not only proliferation (as reported by Diebold *et al*) but also EC survival and EC functions are affected when OxPhos is blocked (Fig. 3 and Extended Data Fig. 3). Adding to the controversy, another report predating Diebolds work had also indicated that the inhibition of mitochondrial OxPhos with oligomycin in HUVEC did *not* interfere with EC sprouting⁴ however, it is important to realise that the same group later revised their initial conclusion in a follow-up study, stating that mitochondrial defects after all *do* affect EC proliferation and survival in spheroid assays⁵. The discrepancy was explained by the use of different experimental setups and timing of inhibitor treatment. These observations clearly indicated that the results obtained with chemical inhibitors and cell culture systems should be considered with caution. We now briefly discuss this issue in our revised manuscript. Our data additionally highlighted the symbiotic metabolic interaction between tumour cells producing lactate and tumour vasculature that metabolise lactate by mitochondrial OxPhos. These results (new Fig. 5) are further substantiated in the revised manuscript and suggest that this symbiotic interaction could be potentially exploited therapeutically to target cancer.

Additional comments:

1. Validation of the KO mice is lacking and is absolutely necessary in light of some of the negative results: evidence of neither efficiency nor specificity of their KO in neither Tie2Cre nor EndSCLCreERT line is provided. It is critical to prove that the phenotypes that the authors observed in the mice are due to the deletion in ECs and not due to the deletion in other cell types such as bone marrow cells (common with Tie2Cre, for example). In addition, confirming efficient KO would be especially critical if the authors want to claim dispensableness of a gene of interest. No data on validation of Cox10 knockout in ECs are provided (eg. qPCR, WB, IHC analysis of Cox10 in the WT vs KO mice).

Response to point 1

We agree with these comments and in the revised manuscript we now include a verification of the *cox10*-KO in ECs by PCR analysis of *cox10*^{EC-/-} mice (Extended Data Fig. 1f), of *cox10*^{ECKO} mice (Extended Data Fig. 3k, Extended Data Fig. 4b, Extended Data Fig. 4c) and of cultured murine ECs (Extended Data Fig. 2a-d). We further include data from mice lacking COX10 in macrophages (*cox10*^{MKO}) and evaluated the KO by PCR (Extended Data Fig. 1g-h).

2. The migration defect that the authors observed could be attributed by the defect in viability. The authors should provide data that viability is intact at the end of migration experiments or remove the claim on the defect in migration. The viability data that the authors provide in the current manuscript is only for a short period of time (15 hrs) while the migration assay was conducted for a longer period of time (48 hrs).

Response to point 2

We thank this reviewer for his/her discerning view. In the revised manuscript we include additional detailed analyses addressing the viability of ECs. These data show that cell death occurs within 24 hrs at 0 mM glucose (means that no glucose supplementation was carried out but serum was not dialysed and cells may be exposed to residual glucose far below culture conditions) exclusively in the OxPhos deficient cells (Fig. 3a and Extended Data Fig. 3a-b). This effect could be rescued by supplementation of physiological glucose levels. While the reduction in viability could be reversed by glucose availability, the observed reduction in migration and proliferation were still observable at normal (5.5 mM, 11 mM) and high (22 mM) glucose concentrations (Fig. 3c and Ext. Data Fig. 3g).

3. The method section needs to be revised to include significant details. The method on tumor necrosis assay (Fig 4k-l), for example, is not provided in the manuscript.

Response to point 3

We apologize for this oversight and have revised the method section accordingly.

Reviewer #3 (Remarks to the Author):

General:

Angiogenesis or the formation of new blood vessels from pre-existing ones, is crucial for development and wound healing but is also involved in tumor vascularization. During angiogenesis, endothelial cells (ECs, lining the lumen of blood vessels) start proliferating and migrating while undergoing a metabolic switch, characterized for example by an increased glycolysis rate to fulfill the increased energy demand. The role of mitochondrial oxidative phosphorylation (OxPhos) in angiogenesis remains debated. In this manuscript, Schiffmann et al. describe that EC-specific deletion of *cox10*, causing an EC-restricted OxPhos deficiency, leads to embryonic lethality but does not affect the quiescent vasculature in healthy adult mice. However, in mouse tumor models, EC-specific *cox10* knockout reduces vascularization and tumor growth. Mechanistically, the authors propose that during angiogenesis, proliferative ECs in a glucose-deprived environment require OxPhos for energy production for which they use lactate coming from hyper-glycolytic tumor cells. While possibly of interest to the broad readership of this journal, the manuscript suffers from a number of shortcomings and overstatements. The following comments will need to be addressed.

We thank this reviewer for his/her substantial effort and very detailed evaluation of our work and would like to respond in detail to his/her specific concerns in the following.

General/major comments:

1. A first major concern pertains to the novelty of the presented work. A crucial role for endothelial OxPhos in developmental and pathological angiogenesis has recently been described by Diebold et al. (2019). The authors thus obviously overstate their case when writing upfront in the abstract: “Our data provide the first genetic evidence for a requirement of mitochondrial respiration for neoangiogenesis during development and disease.” In mice with an EC-specific knockout of complex III subunit Uqcrcq, Diebold et al. showed that OxPhos is required for postnatal retinal and lung vascularization and tumor angiogenesis. They showed the underlying mechanism to be a decreased NAD⁺/NADH ratio which inhibits the TCA cycle flux and thus production of amino acids, and finally impairs EC proliferation but not migration. Did the authors make similar observations for their EC-specific Cox10 knockout? Evaluating one or two key findings from the Diebold paper in the EC-specific Cox10 KO setting, would offer the readers a most interesting view on (dis)similarities between deletion of different OxPhos components in ECs.

Response to point 1

Please see response to reviewer #2 (above, general concern). In our revised manuscript we acknowledge the important study by Diebold *et al*, and discuss the potential impact of our findings with reference to this work.

2. Along the same lines, the authors can further increase novelty of their work by establishing the cox10-deficient embryonic phenotype in more detail (this was left unstudied in the Diebold paper). For instance, different parameters of the vasculature can be studied in whole-mount embryos (rather than in yolk sacs), and physiological parameters like heartbeat can be assessed to obtain insights into overall development of the cardiovascular system.

Response to point 2

We thank this reviewer for his/her suggestions. Addressing these issues, in the revised manuscript we now show detailed analyses of embryonic vascular development by using iDISCO (immunolabeling-enabled three-dimensional imaging of solvent-cleared organs)⁶ combined with fluorescence labelling of endothelial cells after tissue-clearing and confocal microscopy⁷. The data obtained clearly showed that cox10 ablation causes marked vascular alteration. Wild type animals exhibited a detailed and clearly hierarchical blood vessel structure across the whole embryonic tissue, while the EC specific *cox10*-knockout embryos showed heavily reduced CD31 staining and structures indicating failure to maintain the vasculature or induce neoangiogenesis (Fig. 1f and Supplemented Movies 1-6).

3. The authors claim that OxPhos is required to meet the energy demands of ECs but show no hard evidence for this statement, e.g. by measuring intracellular ATP levels or energy charge in *cox10*^{-/-} ECs. Moreover, they did not formally exclude that the underlying mechanism is a defect in TCA cycle (Diebold et al. 2019) or pyrimidine synthesis.

Response to point 3

As suggested, we complemented our study of *cox10*^{KO} ECs with detailed metabolomics analyses and incorporated the results in Fig. 2j. Our results clearly show that in *cox10*^{KO} ECs ATP levels are markedly decreased. Our data by no means formally exclude the role of TCA alteration and clearly showed accumulation of TCA intermediates (other than in Diebold paper).

4. The choice for the high glucose level used in the in vitro assays needs better justification; 22 mM is not simply high but is almost hyperglycemic. The data from the different assays using these high glucose levels would strongly benefit from inclusion of a condition representing more physiologically relevant glucose levels. For the low or zero glucose conditions, was dialyzed serum used in the medium to avoid that remnant levels of glucose coming from the serum are present in the medium and are not accounted for?

Response to point 4

We agree that this is an important issue which was somewhat incompletely described in the original manuscript. The low or zero glucose condition actually means that no glucose supplementation was carried out. Serum was not dialysed and cells may be exposed to residual glucose far below culture conditions (therefore referred to as “low glucose”). In the revised manuscript we now include detailed EC responses (viability, proliferation, migration and sprouting) to gradually decreasing glucose concentrations as was suggested by the reviewer. The new data are now incorporated in Fig. 2-3 and Extended Data Fig. 2-3.

5. In their embryological studies, the authors used the Tie2-Cre transgenic mice to drive EC-specific Cre expression. As Tie2-Cre is not only specific for ECs but also hits the hematopoietic line, it is strongly advisable to confirm key findings with a more specific Cre driver line (e.g. the EndSLC-CreERT model used for the adult mouse studies or other well established EC-specific Cre drivers, e.g.: Flk:Cre (PMID: 14745955) or VE-Cadh:Cre (PMID: 16450386). At the very least, the limitation of using the Tie2-Cre should be clearly acknowledged.

Response to point 5

The reviewer rightly points out that Tie2 in addition to its expression in ECs, is also expressed in hematopoietic cells and that the Tie2-Cre system also hits the hematopoietic lineage^{8,9}. Thus embryonic lethality (E12.5) observed in *cox10*^{EC-/-} embryos following Tie2-Cre driven COX-KO may not only be caused by mitochondrial respiratory dysfunction in ECs but could potentially involve dysfunctions in hematopoietic cells including the lymphocytic - and myeloid lineages. However, previous studies have shown that genetic ablation of mitochondrial respiration in T- and B-cells did not result in embryonic lethality¹⁰⁻¹². Furthermore, in order to address this important issue we bred mice lacking *cox10* in the myeloid compartment by using myeloid-specific LysM-Cre transgenic mice. The data obtained did not reveal embryonic lethality upon specific ablation of the *cox10* gene in the myeloid compartment (Extended Data Fig. 1g-h). Similarly, a recent study using Vav-iCre mice expressing Cre mainly in hematopoietic stem cells (HSCs) but also in ECs showed that alteration of mitochondrial respiration occurred with normal Mendelian ratios at embryonic day 15.5 (E15.5) and only resulted in embryonic lethality at later stages after day E18.5¹³. Together these observations indicate that it is unlikely that non-endothelial tissue dysfunction plays a major part in the embryonic lethality (E12.5) of *cox10*^{EC-/-} mice. We include the new data in our revised manuscript (Extended Data Fig. 1g-h) but also indicate where these results should be interpreted with caution.

6. According to the main text, no obvious phenotypic defects in *cox10*-deficient ECs were observed in the *in vitro* studies. The authors should show supporting data, investigating for instance EC apoptosis, proliferation, senescence or oxidative stress. This also contradicts the authors' statement on the 'incapacity of *cox10*-deficient ECs to proliferate in culture', which is not in line with the observation that OxPhos is only crucial under glucose-limitation. This is all rather confusing, raises questions about experimental conditions and needs further explanation?

Response to point 6

We apologize for the misleading description of the *in vitro* phenotype concerning cellular proliferation or death in COX10-deficient ECs. We observe a marked decrease in cellular proliferation (but not cell death) indicating that COX10-deficient cells have a lower proliferative capacity than COX proficient ECs which is in line with the observations made by Diebold *et al.* We could not detect a relevant amount of cell death under normal glucose concentration (Fig. 3a and Extended Data Fig 3). New data using different concentration of glucose clearly demonstrate that the lack of mitochondrial respiration causes cell death specifically under “low glucose” (zero) concentrations (Fig. 3a and Extended Data Fig. 3a-b) which could not be blocked by caspase inhibitors indirectly indicating that apoptosis is not the major driver of cell death.

7. Fig. 2f-i: The authors showed that low glucose conditions significantly reduce the viability of *cox10* deficient ECs, yet they use these low glucose conditions in scratch wound migration and spheroid assay. How can the authors make sure that the alleged migration defect observed in these assays is not simply due to reduced viability/proliferation of the *cox10* deficient ECs? Also, additional assays specifically assessing EC proliferation or migration (e.g. Boyden chamber) should be performed to be able to make a more sound conclusion on this matter. This is especially relevant, since this statement contrasts with observations made by Diebold et al. who noted a proliferation but no migration defect in OxPhos-deficient ECs.

Response to point 7

In the revised manuscript we now include detailed analyses of viability, proliferation and migration at different glucose concentrations in HUVECs and mouse wt and COX10-deficient ECs (Fig. 3a-e and Ext. Data Fig 3a-g). In particular, we show that the viability of COX10-deficient ECs is not compromised under the glucose concentrations used in the migration experiment (see also response to reviewer #2).

8. The paper would largely benefit from additional analyses of blood vessel parameters in the models for physiological and pathological angiogenesis, e.g. metastasis index for tumor models, hypoxia, vessel maturation (pericyte coverage), vessel perfusion/ leakage (Dextran) or cell proliferation (e.g. Ki67).

Response to point 8

Addressing the reviewers' suggestions, we have conducted additional animal experiments to substantially improve the characterization of tumor angiogenesis in tumour bearing mice with *cox10*-competent and *cox10*-deficient ECs. In the revised manuscript we included new data on i) hypoxia, ii) pericyte coverage, iii) vessel leakage / perfusion and iiiii) metastasis in tumour bearing mice with *cox10*-competent and *cox10*-deficient ECs. Please notice additional Figures 4m-q and Ext. Data Fig. 4f, g and j, k included in the revised manuscript.

9. Fig. 4o-q: These figure panels make little sense without inclusion of a control condition (0 mM lactate added) for both genotypes.

Response to point 9

We thank the reviewer for his comment and have now included the requested control condition (Fig. 5g, h)

Minor comments:

1. From the methods section, it is clear that the OCR and ECAR data were normalized to total protein content (BCA). It is then surprising to see that the data in the graphs are expressed as normalized to arbitrary units (a.u.). Can the authors express the OCR and ECAR data per microgram or milligram protein – which is a much more widely accepted way of presenting these data and which allows for comparison of OCR and ECAR values between independent studies.

Response to minor point 1

As requested, we have repeated the seahorse experiments which now depict OCR and ECAR in the revised Figures 2a-l as values normalized to protein content.

2. Fig. 1c: The authors should present quantifications for the number of blind ends and irregular vessel dilatations in the yolk sacs, rather than simply stating in the main text that the occurrence of these two features was increased in the KOs. The reader might also benefit from an explanation of the yellow and red arrows in the figure legend.

Response to minor point 2

Quantification of blind ends is now included in the revised manuscript (Fig. 1e). There was a trend but no significant differences regarding irregular vessel dilatations, thus we excluded this statement from the revised manuscript.

3. Fig. 1e-i: The authors should present data on the purity of EC populations isolated from COX10fl/fl mice which were used for the in vitro studies.

Response to minor point 3

This information is now included in the revised manuscript (Extended Data Fig. 2c-d).

4. Ext. Fig. 1c: The authors claim that also the vasculature in E10.5 yolk sacs presents severe malformations. However, these are not at all obvious in the representative images. Therefore, quantifications (vascular density, number of blind ends, number of dilatations...) will need to be added to these figure panels in order to make a convincing case here.

Response to minor point 4

Yolk sac vascularisation is now quantified in detail for E12.5 in Fig. 1c-e. No quantitative statements regarding yolk sac vasculature at earlier stages are made in the revised manuscript.

5. Ext. Fig. 1e: The exact same data (except for the heterozygous condition) is presented in Fig. 1d. To avoid all possible confusion, it might be best to explicitly acknowledge this in the legend.

Response to minor point 5

This information is now included in the revised manuscript within the respective Figure legend.

6. Fig. 2: The method section lacks a clear description of the induction of endothelial quiescence, only a reference is mentioned.

Response to minor point 6

This information is now included in the revised manuscript

7. Fig. 2a: Please state to which control condition the drop/gain of OCR and ECAR is compared.

Response to minor point 7

This information is now included in the revised manuscript within the methods section.

8. Fig. 2a: The authors state in the main text “Thus, under saturating amounts of glucose, OxPhos in COX10-competent ECs was suppressed and they metabolically behaved like COX10-deficient ECs.” This should be confirmed by presenting the OCR and ECAR measurements of COX10 competent and deficient ECs exposed to the different glucose concentrations in the same experiment, side by side.

Response to minor point 8

This statement is rephrased in a revised manuscript. Additional new data obtained in mouse ECs and HUVECs clearly show that increased glucose concentration reduce OCR (Fig. 2g-l and Extended Data Fig. 2e).

9. Fig. 2b,d,e,f,g,h,i: Besides analyzing ECs exposed to high and low glucose concentrations, the assays should also be performed in normal glucose conditions (see main comment above).

Response to minor point 9

Our data are further complemented by using different concentrations of glucose. This information is now included in the revised manuscript. Please see Figure 3 a-e and Extended Data Figure 3.

10. Fig. 2f: Did the authors use low glucose conditions (as noted in the figure) or no glucose (as noted in material & methods)? Typographical errors are present in the y-axis legend (the same holds true for Fig. 2h).

Response to minor point 10

Please see response to major point 4

11. Fig. 3: Efficient recombination of the floxed alleles will also need to be shown for the EndSCL-CreERT driver.

Response to minor point 11

This information is now included in the revised manuscript (Extended Data Figure 3k)

12. Fig. 3f: Please adjust the axis title (“junctions” versus “vascular nodal points” in legend).

Response to minor point 12

This title has been adjusted.

13. Ext. Fig. 3: Quantifications of the vascular area in all presented organs will need to be added. The authors should consider using colored images for clarity.

Response to minor point 13

Quantification of all presented organs are now included in the revised manuscript. We considered coloured images, but found that this did not improve clarity. Please see Extended Data Fig. 3l.)

14. Fig. 4c: To get a better understanding of possible differences in the dynamics of wound healing between genotypes, the authors could assess additional parameters besides the total percentage of animals with closed wounds, like for instance evolution of the wound diameter over time.

Response to minor point 14

We regret that the number of animals available for wound closure experiments was limited and precluded a depiction of the evolution of the wound diameter over time. The reviewer may appreciate that further in-depth analysis would require amendment of the experimental permit which was not possible in the time frame of this revision.

Using the wound material available, we evaluated desmin expression as a marker for pericyte coverage, but found no significant differences (Extended Data Fig. 4).

15. Fig. 4h: Statistical analysis needs to be added – the tumor volumes between the two genotypes seem clearly different, yet no statistical analysis is provided.

Response to minor point 15

We apologize for this oversight. This information is now included in the revised manuscript

16. Fig. 4m: OCR data of cox10-deficient ECs with or without lactate supplementation will need to be added.

Response to minor point 16

As cox10-deficient ECs die after about 36 hours of culturing in quiescence medium to induce vascular quiescence before conducting the OCR measurements in then activated ECs, we were not able to include this control group. Nevertheless, the revised Figure 5c-d clearly shows that ECs increase their respiratory rate in the presence of lactate.

17. Fig. 4o,q: Please use uniform output parameters for sprouting and aortic ring analysis (e.g. sprout number/length, extra-aortal GFP+ area; in the same way as presented in figures 2 and 3).

Response to minor point 17

This information is now included in the revised manuscript

18. Ext. Fig. 4a,b: Please enlarge the pictures as well as the asterisks, for clarity.

Response to minor point 18

Figure 4a,b is now in the revised manuscript Figure 4 b,d and has been enlarged and asterisks were added for clarity as requested.

19. Ext. Fig. 4c: This analysis would benefit from measuring the actual lactate levels in the medium of the different cell types.

Response to minor point 19

As requested we measured lactate concentration and incorporated this information in the revised manuscript - please see Fig. 5b.

20. Legends: Please include the sample size.

Response to minor point 20

This information is now included in the revised manuscript.

References:

1. Jabs, M., Rose, A. J., Lehmann, L. H., Taylor, J., Moll, I., Sijmonsma, T. P., Herberich, S. E., Sauer, S. W., Poschet, G., Federico, G., Mogler, C., Weis, E.-M., Augustin, H. G., Yan, M., Gretz, N., Schmid, R. M., Adams, R. H., Gröne, H.-J., Hell, R., Okun, J. G., Backs, J., Nawroth, P. P., Herzig, S. & Fischer, A. Inhibition of Endothelial Notch Signaling Impairs Fatty Acid Transport and Leads to Metabolic and Vascular Remodeling of the Adult Heart. *Circulation* **137**, 2592–2608 (2018).
2. Wunderlich, C. M., Ackermann, P. J., Ostermann, A. L., Adams-Quack, P., Vogt, M. C., Tran, M.-L., Nikolajev, A., Waisman, A., Garbers, C., Theurich, S., Mauer, J., Hövelmeyer, N. & Wunderlich, F. T. Obesity exacerbates colitis-associated cancer via IL-6-regulated macrophage polarisation and CCL- 20/CCR-6-mediated lymphocyte recruitment. *Nature Communications* 1–16 (2018). doi:10.1038/s41467-018-03773-0
3. Diebold, L. P., Gil, H. J., Gao, P., Martinez, C. A., Weinberg, S. E. & Chandel, N. S. Mitochondrial complex III is necessary for endothelial cell proliferation during angiogenesis. *Nature Metabolism* 1–18 (2019). doi:10.1038/s42255-018-0011-x
4. De Bock, K., Georgiadou, M., Schoors, S., Kuchnio, A., Wong, B. W., Cantelmo, A. R., Quaegebeur, A., Ghesquière, B., Cauwenberghs, S., Eelen, G., Phng, L.-K., Betz, I., Tembuysen, B., Brepoels, K., Welti, J., Geudens, I., Segura, I., Cruys, B., Bifari, F., Decimo, I., Blanco, R., Wyns, S., Vangindertael, J., Rocha, S., Collins, R. T., Munck, S., Daelemans, D., Imamura, H., Devlieger, R., Rider, M., Van Veldhoven, P. P., Schuit, F., Bartrons, R., Hofkens, J., Fraisl, P., Telang, S., DeBerardinis, R. J., Schoonjans, L., Vinckier, S., Chesney, J., Gerhardt, H., Dewerchin, M. & Carmeliet, P. Role of PFKFB3-driven glycolysis in vessel sprouting. *Cell* **154**, 651–663 (2013).
5. Vandekerke, S., Dubois, C., Kalucka, J., Sullivan, M. R., García-Caballero, M., Goveia, J., Chen, R., Diehl, F. F., Bar-Lev, L., Souffreau, J., Pircher, A., Kumar, S., Vinckier, S., Hirabayashi, Y., Furuya, S., Schoonjans, L., Eelen, G., Ghesquière, B., Keshet, E., Li, X., Vander Heiden, M. G., Dewerchin, M. & Carmeliet, P. Serine Synthesis via PHGDH Is Essential for Heme Production in Endothelial Cells. *Cell Metabolism* 1–29 (2018). doi:10.1016/j.cmet.2018.06.009

6. Renier, N., Wu, Z., Simon, D. J., Yang, J., Ariel, P. & Tessier-Lavigne, M. iDISCO: A Simple, Rapid Method to Immunolabel Large Tissue Samples for Volume Imaging. *Cell* **159**, 896–910 (2014).
7. Klingberg, A., Hasenberg, A., Ludwig-Portugall, I., Medyukhina, A., Männ, L., Brenzel, A., Engel, D. R., Figge, M. T., Kurts, C. & Gunzer, M. Fully Automated Evaluation of Total Glomerular Number and Capillary Tuft Size in Nephritic Kidneys Using Lightsheet Microscopy. *JASN* **28**, 452–459 (2017).
8. Constien, R., Forde, A., Liliensiek, B., Grone, H. J., Nawroth, P., Hammerling, G. & Arnold, B. Characterization of a novel EGFP reporter mouse to monitor Cre recombination as demonstrated by a Tie2 Cre mouse line. *genesis* **30**, 36–44 (2001).
9. Tang, Y., Harrington, A., Yang, X., Friesel, R. E. & Liaw, L. The contribution of the Tie2+ lineage to primitive and definitive hematopoietic cells. *genesis* **48**, 563–567 (2010).
10. Weinberg, S. E., Singer, B. D., Steinert, E. M., Martinez, C. A., Mehta, M. M., Martínez-Reyes, I., Gao, P., Helmin, K. A., Abdala-Valencia, H., Sena, L. A., Schumacker, P. T., Turka, L. A. & Chandel, N. S. Mitochondrial complex III is essential for suppressive function of regulatory T cells. *Nature* 1–28 (2019). doi:10.1038/s41586-018-0846-z
11. Cabon, L., n-Malo, P. G. A., Bouharrou, A., e, L. D. E., Brunelle-Navas, M.-N., Lorenzo, H. K., Gross, A. & Susin, S. A. BID regulates AIF-mediated caspase-independent necroptosis by promoting BAX activation. *Cell Death and Differentiation* 1–12 (2019). doi:10.1038/cdd.2011.91
12. Bertaux, A., Cabon, L., Brunelle-Navas, M.-N., Bouchet, S., Nemazanyy, I. & Susin, S. A. Mitochondrial OXPHOS influences immune cell fate: lessons from hematopoietic AIF- deficient and NDUFS4-deficient mouse models. *Cell Death and Disease* 1–4 (2018). doi:10.1038/s41419-018-0583-0
13. Ansó, E., Weinberg, S. E., Diebold, L. P., Thompson, B. J., Malinge, S., Schumacker, P. T., Liu, X., Zhang, Y., Shao, Z., Steadman, M., Marsh, K. M., Xu, J., Crispino, J. D. & Chandel, N. S. The mitochondrial respiratory chain is essential for haematopoietic stem cell function. *Nat Cell Biol* **19**, 614–625 (2017).

Reviewers' comments:

Reviewer #1 (Remarks to the Author):

I found this paper an interesting contribution on the role of OxPhos in Endothelial Cells physiology and disease, particularly in angiogenesis associated with wound healing and tumour biogenesis. The paper has improved following the suggestions of the reviewers, with the exception of mine that were perhaps too far beyond the scope of the work.

Reviewer #2 (Remarks to the Author):

The authors have addressed most comments. Novelty and mechanistic insights remain weak, however. Moreover, the validation of EC specificity and efficient knockout of their in vivo system remains unsatisfactory. The authors only provide genotyping data by PCR from whole tail which answers neither of these questions. They also provide a WB data of COX10 after Cre treatment in mouse ECs with floxed alleles, but these data are obtained by in vitro treatment of Cre protein to the cells. They don't show it in vivo. Isolation of ECs followed by WB or qPCR is critical to demonstrate efficiency. And isolation of other cells, especially bone marrow, is critical to demonstrate specificity.

Reviewer #3 (Remarks to the Author):

General:

By performing additional in vitro and in vivo analyses, the authors have extensively revised and thereby strengthened their manuscript entitled "Mitochondrial respiration controls neoangiogenesis during wound healing and tumour growth". However, some of the comments raised were not entirely or properly addressed. In the reviewer's opinion, still a number of key issues remain unresolved and should be addressed prior to considering the manuscript for publication.

General/major comments:

1. Reply to comment 2: While this reviewer certainly appreciates the inclusion of supplemental movies illustrating the vascularization defect in whole embryos upon Cox10 knockout, no legends or information specifying which movie refers to which condition, are present in the revised manuscript. Furthermore, the authors state in their rebuttal that iDISCO was used to visualize the vasculature in whole embryos, yet this information seems to have gone missing from the methods section. Finally, the representative images deriving from these CD31 whole embryo stainings are now included as Figure 1f but accompanying quantifications are missing. These should be provided.

2. Reply to comment 4: The authors made a great effort in complementing their results with additional glucose concentration conditions. The reviewer appreciates the details in the rebuttal about how low glucose conditions were achieved (i.e. serum was not dialysed and cells may be exposed to residual glucose far below culture conditions). However, these details seem not to have made it into the manuscript - the text now simply mentions "low glucose" without further explanation. The authors should provide the necessary details on these low glucose conditions in the methods section. Finally, Figure 3a on cell death at different glucose concentrations is overcrowded with symbols (which mostly overlap in the bottom part of the graph) for the different conditions which makes the figure difficult to digest. Can a more intelligible way of representing these data be found?

3. Reply to comment 7: Figure 3b-e: While the reviewer appreciates the authors' efforts to assess viability of the COX10-deficient ECs at the glucose concentrations used in the migration assays, it remains a rather indirect way of proving that the migration defect is not merely a reflection of reduced proliferation. The golden standard to disentangle a proliferation defect from a pure migratory phenotype is to pre-treat the cells with mitomycin C prior to performing the scratch assay. Similarly, spheroid sprouting assays can include mitomycin C treatment to judge the impact of proliferation versus migration on the observed sprouting defect. Given that in their revised version, the authors nowhere mention the use of mitomycin C in these assays, this reviewer recommends to include mitomycin C pretreatment in one of the assays to make a better call on the migration defect (and it being possibly confounded by proliferation).

4. Reply to comment 8: The reviewer appreciates the new experiments performed by the authors to analyze additional blood vessel parameters in pathological conditions. Quantification methods, and particularly for the Dextran leakage and 'vessel bound Dextran' (as is written in the figure legend), should be further specified in the methods section. This reviewer also has a concern about Figure 4n-o, where, according to the figure legend, only 3 animals per genotype were used. While the differences came out as significant, there is quite an important variability between the 3 animals. More confidence could be put in these findings by increasing the group sizes. However, the reviewer leaves it to the Editor to decide whether this is required or not.

5. The current version of the manuscript does no longer contain results comparing proliferative versus quiescent ECs (previously Figure 2d-e). Still the methods section contains information on how vascular quiescence was achieved. The reviewer has 2 comments on this: (i) What was the rationale for removing these data? (ii) If data on quiescent ECs are included in the manuscript, can the authors cite a reference or provide supporting data that the protocol they used efficiently induces quiescence since it differs from other established methods such as contact inhibition and Dll4-induced quiescence. The authors' response to minor point 16 also refers to quiescence. It is overall confusing; were OCR measurements (always) performed in the so-called quiescence medium? Can the authors please clarify?

6. The statement that supra-physiological glucose concentrations can partially rescue migration and sprouting in COX10KO cells is not supported by the data (Figure 3b-e). No rescue can be observed in the scratch wound assay, and there is no statistical validation of the increase in sprouting between COX10KO in 22 versus 11 mM glucose.

Minor comments:

1. Response to minor comment 1: Representing OCR and ECAR data as normalized to amount of protein is still not consistently done throughout the manuscript. As mentioned in the methods, data were either normalized to microgram of protein, which is correct but not adequately reported in the y axis of Figure 2a-l, or normalized to optical density (OD) obtained after BCA. The latter, although indeed being a proxy for protein levels, does not indicate the actual protein levels and thus does not permit meaningful comparison between independent studies. OD should be replaced by amount of protein as in Figure 2.

2. Response to minor comment 3: EC purity is now assessed by testing several endothelial or non-endothelial markers by QPCR. However, this does not provide information on the real, percentual purity. Instead, cytometry analysis would be highly recommended.

3. Response to minor comment 11: Additional details about the genotyping strategy should be provided. How do the authors explain the low recombination efficiency ("KO") in *cox10^{fl/fl}* EndSCLCreERT mice (Ext Figure 3k) especially in comparison with the recombination efficiency in LysMCre mice (Ext. Figure. 1h)? Which tissue was used for genotyping in Ext. Figure 3k? To truly validate the in vivo excision efficiency, the mRNA/protein level of *cox10* in ECs isolated from tamoxifen-treated mice could be assessed.

4. Response to minor comment 18: The asterisks (as mentioned in the legend) are not visible on the figure.

5. Response to minor comment 19: In Figure 5a-b, the statistics should be better explained. In its current version, the reader cannot understand which conditions were compared. Additionally, the high ECAR values for MAFs do not translate into lactate secretion (where MAFs actually have the lowest amount across the cell types tested). Can the authors comment on this?

6. A typographical error is present on line 615 “decreasing glusose concentrations”

Point-by-Point reviewer response

Reviewer #1 (Remarks to the Author):

I found this paper an interesting contribution on the role of OxPhos in Endothelial Cells physiology and disease, particularly in angiogenesis associated with wound healing and tumour biogenesis. The paper has improved following the suggestions of the reviewers, with the exception of mine that were perhaps too far beyond the scope of the work.

Again, we thank this reviewer for his/her very valuable suggestions to further explore the role of mitochondrial bioenergetics in endothelial cells under challenging metabolic conditions. While the scope of our study was intended to establish the role of mitochondrial respiratory function in ECs during embryonic development and during the course of tumour growth and wound healing, we tried to accommodate the reviewer's suggestions and included further experimental work on the role of mitochondrial respiration in vasculature function under restrictive metabolic conditions. To complement this, we immediately started experiments to establish further metabolic conditions and we obtained the necessary permissions for animal work from the regulatory authorities. While we have managed to initiate some studies during the revision period, we regret that at this point in time the preliminary nature of the results obtained did not yet provide a solid enough basis for conclusive interpretation, which we have already summarised in our previous point-by-point response. We have come to realize that more detailed analyses will require additional expertise and time to properly address these important issues, but this is in our view far beyond the scope of the current manuscript as this reviewer also recognised. We are fully aware of the delicate nature of some of these questions and would like to avoid the potential for misinterpretation based on incomplete analysis. We hope that this reviewer appreciates our concerns and accepts our apology for not being able to address the issues raised by him/her more comprehensively at this time.

Reviewer #2 (Remarks to the Author):

The authors have addressed most comments. Novelty and mechanistic insights remain weak, however. Moreover, the validation of EC specificity and efficient knockout of their in vivo system remains unsatisfactory. The authors only provide genotyping data by PCR from whole tail which answers neither of these questions. They also provide a WB data of COX10 after Cre treatment in mouse ECs with floxed alleles, but these data are obtained by in vitro treatment of Cre protein to the cells. They don't show it in vivo. Isolation of ECs followed by WB or qPCR is critical to demonstrate efficiency. And isolation of other cells, especially bone marrow, is critical to demonstrate specificity.

We appreciate the comments. In order to demonstrate EC specificity, we now include qRT-PCR analyses showing tamoxifen-induced *cox10* knock-out in ECs, but not in bone marrow-derived macrophages (BMDMs), isolated from *cox10^{fl/fl}* or *cox10^{fl/fl} EndSCLCreERT* animals after tamoxifen treatment (please see below in b). The efficacy of tamoxifen in inducing COX10 KO was further verified by Western blotting in *cox10^{fl/fl} EndSCLCreERT* mice (c, upper panel). We included also WB data showing that tamoxifen-induced COX10 knock-out only occurs in *cox10^{fl/fl} EndSCLCreERT* but not in *cox10^{fl/fl}* mice providing a proof of tamoxifen-induced

EndSCLCreERT-dependent *cox10* knock-out in ECs (c, lower panel). These data appear in the new Extended Figure 4b-c.

b

c

Reviewer #3 (Remarks to the Author):

General:

By performing additional in vitro and in vivo analyses, the authors have extensively revised and thereby strengthened their manuscript entitled “Mitochondrial respiration controls neoangiogenesis during wound healing and tumour growth”. However, some of the comments raised were not entirely or properly addressed. In the reviewer’s opinion, still a number of key issues remain unresolved and should be addressed prior to considering the manuscript for publication.

We thank this reviewer again for the detailed assessment of our manuscript.

General/major comments:

1. Reply to comment 2: While this reviewer certainly appreciates the inclusion of supplemental movies illustrating the vascularization defect in whole embryos upon *Cox10* knockout, no legends or information specifying which movie refers to which condition, are present in the revised manuscript. Furthermore, the authors state in their rebuttal that iDISCO was used to visualize the vasculature in whole embryos, yet this information seems to have

gone missing from the methods section. Finally, the representative images deriving from these CD31 whole embryo stainings are now included as Figure 1f but accompanying quantifications are missing. These should be provided.

In the revised manuscript we provide additional information on the videos in the methods section and named the respective files accordingly. The iDISCO analysis is now also included. Finally, as requested, we have now quantified CD31 in the images presented in Figure 1f (compare new Figure 1g).

2. Reply to comment 4: The authors made a great effort in complementing their results with additional glucose concentration conditions. The reviewer appreciates the details in the rebuttal about how low glucose conditions were achieved (i.e. serum was not dialysed and cells may be exposed to residual glucose far below culture conditions). However, these details seem not to have made it into the manuscript - the text now simply mentions “low glucose” without further explanation. The authors should provide the necessary details on these low glucose conditions in the methods section. Finally, Figure 3a on cell death at different glucose concentrations is overcrowded with symbols (which mostly overlap in the bottom part of the graph) for the different conditions which makes the figure difficult to digest. Can a more intelligible way of representing these data be found?

Details on the glucose conditions have now been included in the revised manuscript which can be found in the methods section (subitem ‘Endothelial cells’); we replaced ‘low’ or ‘high’ glucose with the actual concentrations within the text and the figures. Furthermore, we changed the data presentation in Figure 3a and hope that these data are now more intelligible for the readers.

3. Reply to comment 7: Figure 3b-e: While the reviewer appreciates the authors’ efforts to assess viability of the COX10-deficient ECs at the glucose concentrations used in the migration assays, it remains a rather indirect way of proving that the migration defect is not merely a reflection of reduced proliferation. The golden standard to disentangle a proliferation defect from a pure migratory phenotype is to pre-treat the cells with mitomycin C prior to performing the scratch assay. Similarly, spheroid sprouting assays can include mitomycin C treatment to judge the impact of proliferation versus migration on the observed sprouting defect. Given that in their revised version, the authors nowhere mention the use of mitomycin C in these assays, this reviewer recommends to include mitomycin C pre-treatment in one of the assays to make a better call on the migration defect (and it being possibly confounded by proliferation).

We appreciate the reviewer’s suggestion to more clearly distinguish proliferation from pure migration and now include the data derived from additional scratch wound assays in the presence of mitomycin C (new Extended Data Figure 3h). The presence of mitomycin C did not significantly affect the observed differences between *cox10^{fl/fl}* and *cox10^{KO}* ECs ability to close scratch wounds *in vitro* within 16 h. We therefore conclude that reduced cell proliferation was not the main cause of delayed scratch wound closure in our system. However, these experiments cannot formally exclude the possibility that the angiogenesis defects observed *in vivo* also significantly involve reduced EC proliferation.

4. Reply to comment 8: The reviewer appreciates the new experiments performed by the

authors to analyze additional blood vessel parameters in pathological conditions. Quantification methods, and particularly for the Dextran leakage and ‘vessel bound Dextran’ (as is written in the figure legend), should be further specified in the methods section. This reviewer also has a concern about Figure 4n-o, where, according to the figure legend, only 3 animals per genotype were used. While the differences came out as significant, there is quite an important variability between the 3 animals. More confidence could be put in these findings by increasing the group sizes. However, the reviewer leaves it to the Editor to decide whether this is required or not.

We now provide further details on the quantification methods in the revised manuscript (please compare method section sub-item ‘Immunohistochemistry’). After conferring with the editor, we agreed to abstain from further experiments to increase the number of mice to address this particular point.

5. The current version of the manuscript does no longer contain results comparing proliferative versus quiescent ECs (previously Figure 2d-e). Still the methods section contains information on how vascular quiescence was achieved. The reviewer has 2 comments on this: (i) What was the rationale for removing these data? (ii) If data on quiescent ECs are included in the manuscript, can the authors cite a reference or provide supporting data that the protocol they used efficiently induces quiescence since it differs from other established methods such as contact inhibition and Dll4-induced quiescence. The authors’ response to minor point 16 also refers to quiescence. It is overall confusing; were OCR measurements (always) performed in the so-called quiescence medium? Can the authors please clarify?

While we have no doubt concerning the quality of our data comparing proliferative *versus* quiescent ECs, we decided not to include these results in the revised manuscript because similar data have previously been published by our group (Coutelle *et al*, 2014) and in our view are dispensable for the narrative of the current manuscript.

Concerning OCR measurement conditions: All Seahorse experiments were performed in the respective Agilent assay medium (now detailed in the revised manuscript in the methods section). Prior to this, cells were cultured in a 1:1 mixture of EGM2 and full supplemented DMEM. To measure OCR with or without lactate the MitoStress assay was conducted using a modified assay medium lacking pyruvate and glucose. Quiescence (which is no longer included in the manuscript) was induced in confluent ECs by cultivation in basal EC medium depleted of growth factors for 36 h. Verification of quiescence was done morphologically and by assessing CD105 (endoglin) expression (FACS analysis) (Coutelle *et al*, 2014). We adopted the revised manuscript accordingly to avoid any confusion and thank the reviewer for his clarifying comments.

6. The statement that supra-physiological glucose concentrations can partially rescue migration and sprouting in COX10KO cells is not supported by the data (Figure 3b-e). No rescue can be observed in the scratch wound assay, and there is no statistical validation of the increase in sprouting between COX10KO in 22 versus 11 mM glucose.

We agree with the reviewer and apologize for these misinterpretations. In the revised manuscript we now state: ‘Specifically, 3D spheroid sprouting assays and scratch

wound assays showed that sprouting response and migration of *cox10^{KO}* ECs was significantly hampered which could not be rescued by supra-physiological glucose concentrations (Fig. 3b-e).'

Minor comments:

1. Response to minor comment 1: Representing OCR and ECAR data as normalized to amount of protein is still not consistently done throughout the manuscript. As mentioned in the methods, data were either normalized to microgram of protein, which is correct but not adequately reported in the y axis of Figure 2a-I, or normalized to optical density (OD) obtained after BCA. The latter, although indeed being a proxy for protein levels, does not indicate the actual protein levels and thus does not permit meaningful comparison between independent studies. OD should be replaced by amount of protein as in Figure 2.

We appreciate the comments and have corrected the OCR and ECAR data normalized to the amount of protein as requested which is now included in the revised Figures. We also rephrased the method section to clarify this point for the readers.

2. Response to minor comment 3: EC purity is now assessed by testing several endothelial or non-endothelial markers by QPCR. However, this does not provide information on the real, percentual purity. Instead, cytometry analysis would be highly recommended.

Following the reviewer's suggestion, we conducted flow cytometry analysis as requested. Please refer to the new Extended Data Figure 2a for details. Please note that cells for experiments were additionally re-sorted after passage one in order to gain highest possible purity (Please refer to method section on Endothelial Cells).

3. Response to minor comment 11: Additional details about the genotyping strategy should be provided. How do the authors explain the low recombination efficiency ("KO") in *cox10^{fl/fl}* EndSCLCreERT mice (Ext Figure 3k) especially in comparison with the recombination efficiency in LysMCre mice (Ext. Figure. 1h)? Which tissue was used for genotyping in Ext. Figure 3k? To truly validate the in vivo excision efficiency, the mRNA/protein level of *cox10* in ECs isolated from tamoxifen-treated mice could be assessed.

The methods section now includes additional details on the genotyping strategy as requested. The difference observed in the KO band intensity in samples derived from *cox10^{fl/fl}* EndSCLCreERT versus LysMCre mice results from the different tissues that were used to examine the gene recombination. The PCR for examining the *cox10* knock-out in EndSCLCreERT mice after tamoxifen treatment (new Extended Data Fig. 4a) was done on ear biopsies which do not exclusively comprise of ECs but also include other cell types (e.g. skin fibroblasts, keratinocytes that do not undergo gene recombination). This analysis simply shows that tamoxifen treatment induces *cox10* gene ablation but not in every cell type found in the tissue. The PCR analysis for LysMCre-mediated gene recombination in macrophages (Extended Data Fig.1h) was carried out on isolated BMDMs from mice after cultivation in selective medium to enrich a pure myeloid culture.

To show that tamoxifen treatment efficiently and specifically induced *cox10* KO in ECs we include additional WB and RT-PCR analyses showing *cox10* KO in ECs isolated from EndSCLCreERT *cox10^{fl/fl}* mice after tamoxifen treatment (Extended Data Figure 4b and c). Additional RT-PCR analyses of *cox10* in BMDMs isolated from these mice show that tamoxifen treatment selectively induced *cox10* KO in ECs but not in myeloid cells (Extended Data Figure 4b) (please see also response to reviewer #2).

4. Response to minor comment 18: The asterisks (as mentioned in the legend) are not visible on the figure.

This has been corrected.

5. Response to minor comment 19: In Figure 5a-b, the statistics should be better explained. In its current version, the reader cannot understand which conditions were compared. Additionally, the high ECAR values for MAFs do not translate into lactate secretion (where MAFs actually have the lowest amount across the cell types tested). Can the authors comment on this?

We thank this reviewer and have improved the presentation of the statistics as requested. As reported previously, the rate at which cells acidify supernatant does not only correlate with the conversion rate of glucose to lactate⁻ and H⁺ but also involves O₂ consumption, TCA cycle and/or CO₂ production. This substantially differs among tissues, by substrate usage, or by the limitations placed on respiration (Mookerjee SA et al, *Biochim Biophys Acta*. 2015 Feb;1847(2):171-181). This knowledge may serve as a potential explanation for our observations with MAFs showing that the acidification rate measured/calculated by ECAR does not correlate with the amount of lactate in the supernatant, which was measured separately. We agree that in the present form the MAFs data appear to be inconsistent. Since MAFs do not add important information considering the main statement of this figure (mainly comparing lactate production in tumour cells and ECs) we decided to remove the MAFs data from the revised manuscript.

6. A typographical error is present on line 615 “decreasing glusucose concentrations”

This has been corrected.

Reviewers' comments:

Reviewer #2 (Remarks to the Author):

the authors have addressed satisfactorily my comments

Reviewer #3 (Remarks to the Author):

The additional revisions performed by the authors significantly improved the manuscript and most comments have now been sufficiently addressed. However, there are a few minor points remaining unresolved.

1. Line 636: Here, the authors should refer to data presented in Extended Data Fig 2a-c and not, as noted in the text, to 'Extended Data Fig. 3a-c'.
2. Regarding major comment 3 (related to initial comment 7): The authors now included data of scratch wound experiments in cox10fl/fl and in cox10 KO ECs treated with the proliferation inhibitor mitomycin C (2h, 2 μ g/ml) to assess the contribution of a possible proliferation defect to the migratory phenotype. In this Reviewer's opinion, a 2-hour pretreatment with mitomycin C seems rather short. Can the authors provide a reference for this protocol showing that a 2-hour treatment indeed blocks proliferation? Alternatively, if available, the authors might provide own data showing that 2-hour mitomycin C treatment effectively blocks proliferation.
3. Regarding minor comment 1 (related to initial minor comment 1): While the authors now state in the methods section that OCR and ECAR data have been normalized to protein amount, the y-axes of the revised figures have not been adapted accordingly.
4. Regarding minor comment 2 (related to initial minor comment 3): The authors failed to provide data regarding the actual purity of ECs which were used for their experiments. They merely provide FACS data assessing the purity of ECs after a single CD31 enrichment, which is rather low (73%) and less relevant, as it was followed by an additional enrichment step. FACS data assessing the purity of ECs after performing this additional enrichment step should be provided.

Point-By-Point rev3

Reviewer #2 (Remarks to the Author):

the authors have addressed satisfactorily my comments.

We thank this reviewer for his/her constructive evaluation of our manuscript.

Reviewer #3 (Remarks to the Author):

The additional revisions performed by the authors significantly improved the manuscript and most comments have now been sufficiently addressed. However, there are a few minor points remaining unresolved.

1. Line 636: Here, the authors should refer to data presented in Extended Data Fig 2a-c and not, as noted in the text, to ‘Extended Data Fig. 3a-c’.

The text is adopted accordingly (highlighted).

2. Regarding major comment 3 (related to initial comment 7): The authors now included data of scratch wound experiments in *cox10^{fl/fl}* and in *cox10* KO ECs treated with the proliferation inhibitor mitomycin C (2h, 2 μ g/ml) to assess the contribution of a possible proliferation defect to the migratory phenotype. In this Reviewer’s opinion, a 2-hour pretreatment with mitomycin C seems rather short. Can the authors provide a reference for this protocol showing that a 2-hour treatment indeed blocks proliferation? Alternatively, if available, the authors might provide own data showing that 2-hour mitomycin C treatment effectively blocks proliferation.

We designed this specific experimental procedure as described recently by Diebold *et al.* published in *Nat Metabol.* 2019 ¹ which was explicitly recommended to us by the reviewers in the initial revision. Independently, previous data already showed measurable effects after only 6 seconds following Mitomycin C treatment and persistent/irreversible effects after 30 minutes of Mitomycin ² indicating that 2 hours of pre-treatment with Mitomycin C in our studies is sufficient to achieve proliferation defects. As requested, Diebold *et al.*, is additionally cited for the section describing this experimental setup specifically.

3. Regarding minor comment 1 (related to initial minor comment 1): While the authors now state in the methods section that OCR and ECAR data have been normalized to protein amount, the y-axes of the revised figures have not been adapted accordingly.

The description of Y-axes is adopted in the revised manuscript.

4. Regarding minor comment 2 (related to initial minor comment 3): The authors failed to provide data regarding the actual purity of ECs which were used for their experiments. They merely provide FACS data assessing the purity of ECs after a single CD31 enrichment, which is rather low (73%) and less relevant, as it was followed by an additional enrichment step. FACS data assessing the purity of ECs after performing this additional enrichment step should be provided.

As requested, the FACS data from a representative experiment are now incorporated in the revised manuscript and appear in Ext. Data Fig. 2a. The purity was further improved to around 91% based on CD31 staining after the second sorting round.

References:

1. Diebold, L. P., Gil, H. J., Gao, P., Martinez, C. A., Weinberg, S. E. & Chandel, N. S. Mitochondrial complex III is necessary for endothelial cell proliferation during angiogenesis. *Nat Metab* **1**, 158–171 (2019).
2. Roh, D. S., Cook, A. L., Rhee, S. S., Joshi, A., Kowalski, R., Dhaliwal, D. K. & Funderburgh, J. L. DNA Cross-linking, Double-Strand Breaks, and Apoptosis in Corneal Endothelial Cells after a Single Exposure to Mitomycin C. *Investigative Ophthalmology & Visual Science* **49**, 4837–4843 (2008).